# Exon-junction complex association with stalled ribosomes and slow translation-independent disassembly

Olivier Bensaude [1,3,4] ✉, Isabelle Barbosa[1,3], Lucia Morillo[1], Rivka Dikstein[2] & Hervé Le Hir[1,4] ✉

Exon junction complexes are deposited at exon-exon junctions during splicing. They are primarily known to activate non-sense mediated degradation of transcripts harbouring premature stop codons before the last intron. According to a popular model, exon-junction complexes accompany mRNAs to the cytoplasm where the first translating ribosome pushes them out. However, they are also removed by uncharacterized, translation-independent mechanisms. Little is known about kinetic and transcript specificity of these processes. Here we tag core subunits of exon-junction complexes with complementary split nanoluciferase fragments to obtain sensitive and quantitative assays for complex formation. Unexpectedly, exon-junction complexes form large stable mRNPs containing stalled ribosomes. Complex assembly and disassembly rates are determined after an arrest in transcription and/or translation. 85% of newly deposited exon-junction complexes are disassembled by a translation-dependent mechanism. However as this process is much faster than the translation-independent one, only 30% of the exon-junction complexes present in cells at steady state require translation for disassembly. Deep RNA sequencing shows a bias of exon-junction complex bound transcripts towards microtubule and centrosome coding ones and demonstrate that the lifetimes of exon-junction complexes are transcript-specific. This study provides a dynamic vision of exon-junction complexes and uncovers their unexpected stable association with ribosomes.

Exon-junction complexes (EJC) are multiprotein complexes found in eukaryotes, from budding yeasts to plants and mammals[1–4]. They comprise a core of three proteins, eIF4A3/DDX48, MAGOH, and Y14/RBM8A interacting with several accessory partners, such as MLN51/CASC3/BARENTSZ[5], the ASAP or the PSAP complexes comprising RNPS1, SAP18 and ACIN1 or PNN proteins, respectively[6]. EJCs are primarily known to activate non-sense mediated degradation (NMD) of transcripts harbouring premature stop codons (PTC) before the last intron[7–9]. This process contributes to mRNA quality control.

Independently of NMD, EJCs influence alternative splicing, provide a binding platform for factors involved in transcript nuclear export, enhance translation efficiency and repress m6A methylation of mRNAs[3,10,11]. Furthermore, stable EJC binding is required for the localization of specific transcripts. *Oskar* in Drosophila oocytes and *NIN* in mammalian cells are the only ones that have been described so far[12,13].

EJCs are assembled on exon-exon junctions by the spliceosome, an RNA/protein structure that undergoes extensive changes in snRNA

[1]Institut de Biologie de l'Ecole Normale Supérieure (IBENS), Ecole Normale Supérieure, CNRS, INSERM, PSL Research University, Paris, France. [2]Department of Biomolecular Sciences, The Weizmann Institute of Science, Rehovot, Israel. [3]These authors contributed equally: Olivier Bensaude, Isabelle Barbosa. [4]These authors jointly supervised this work: Olivier Bensaude, Hervé Le Hir. ✉e-mail: bensaude@bio.ens.psl.eu; lehir@bio.ens.psl.eu

**Fig. 1 | Luciferase activity of an EJC-NanoBiT. a** The 3D structure of an in vitro assembled EJC PDB2J0S[24] shows that MAGOH N-terminus is close to Y14 C-terminus and that Magoh C-terminus is close to eIF4A3 C-terminus; **b** MAGOH stably binds Y14, SmBiT fused to C-terminus (C) of Y14 interacts with LgBiT fused to N-terminus (N) of MAGOH thereby recapitulating NanoBiT luciferase activity; **c** MAGOH and eIF4A3 interact only when belonging to an assembled EJC, luciferase activity is recapitulated if LgBiT and SmBiT are fused to their respective C-termini (C); **d**, **e** relative luciferase activities (RLUs) obtained following transient transfection in

HEK 293 cells; **f**, **g** HEK293 derived cell lines stably expressing eIF4A3-SmBiT (OB2 cells) or MAGOH-LgBiT (OB3 cells) are transiently transfected with plasmids expressing either wild-type or mutant HA-MAGOH-LgBiT or Flag-eIF4A3-SmBiT; data for **d**–**g** are from at least two biological replicates. Western blots probe the expression of tagged transfected proteins, GAPDH is a control for equal amount of cells loaded in every lane. GAPDH, eIF4A3, MAGOH and Y14 proteins are detected on Western blots with specific antibodies diluted 1/1000. Source data are provided in the Source Data file.

and protein composition following a complex choreography[14]. The cleavage of the 5′ exon-intron junction occurs within the catalytically active complex, B*. Exons are ligated in the post-spliceosomal complex C*. The spliced transcript and the intron lariat are released while the remaining snRNPs are recycled. Cryogenic electron microscopy data show that eIF4A3 enters the activated spliceosome complex, B$^{act}$, before the formation of complex B*[15]. Next, a MAGOH/Y14 heterodimer would bind to eIF4A3 within complex C just before exon-exon ligation[16]. According to a popular model, EJCs accompany mRNAs along their export to the cytoplasm where the first translating ribosome then pushes them out[17]. When EJCs are not removed, they recruit NMD factors and activate transcript degradation[8,18]. However, EJCs can also be removed by yet to be characterized, translation-independent mechanisms[19] and NMD can occur independently of EJCs[20].

The kinetics of EJC assembly and disassembly from its target transcript are poorly documented. Yet, the rates of both processes dictate the EJC's intracellular abundance. EJCs assembly rates might reflect splicing rates. But how long does it take for EJCs to be removed and disassembled? To address this question we turned onto a NanoBiT split luciferase system[21]. This highly sensitive and quantitative method is well adapted for monitoring protein interaction dynamics because of its fast protein folding and low intrinsic affinity between split fragments.

Here, we show that co-expression of C-terminal NanoBiT fusions of the EJC core subunits Magoh and eIF4A3, recapitulates a strong luciferase activity. Measuring this luciferase activity following arrests in transcription and/or translation, permits to evaluate the relative contributions of translation-dependent and translation-independent disassembly mechanisms. The split-luciferase assay provides an overall view on EJC disassembly rates, but do EJC disassembly rates depend on

the transcript they bind to? To address this question, we used RIPseq to investigate the transcript dependence of EJC disassembly rate. Indeed, we find a wide spectrum of EJC persistence times on their transcript targets, indicating that the lifetime of EJC particles is transcript-specific.

## Results

### Engineering an EJC-NanoBiT

To investigate the EJC disassembly, we used a NanoBiT protein complementation assay[21]. The nanoluciferase cDNA has been split to code for a large (158 a.a.) N-terminal inactive fragment (LgBiT) and a small (11 a.a.) C-terminal peptide (SmBiT). These fragments are fused with a 14 a.a. linker to either the N- or the C-termini of the proteins of interest. Both fragments have been optimized to fold rapidly and display low affinity for each other in order not to enhance association between their partners as observed with other split systems.

The core EJC comprises three proteins, eIF4A3, MAGOH and Y14[22]. Y14 and Magoh form stable heterodimers[23]. Several 3-D structures are available[24,25] (Fig. 1a). As the MAGOH N-terminus is close (circa 15 Å) to Y14 C-terminus, SmBiT and LgBit fragments were fused to the C- and N- termini of MAGOH and Y14, respectively (Fig. 1b). An elevated luciferase activity was detected in lysates from cells cotransfected with both fusions (Fig. 1d, lane 4). In contrast, cotransfection of SmBiT-N-MAGOH and MAGOH-C-LgBiT did not result in any significant activity (Fig. 1d, lane 5). LgBiT and SmBiT fragments have to be brought in a favourable 3-D arrangement, to recapitulate luciferase activity.

Our objective is to detect the luciferase activity only when an EJC is assembled. A contact between eIF4A3 and the Y14/MAGOH pair can be considered as the signature of an assembled EJC. A close look at the

3-D structures shows that eiF4A3 and MAGOH C-termini are close (circa 15 Å) but far away from their N-termini or either Y14 termini (Fig. 1a). Therefore, the SmBiT and LgBiT fragments were fused to the C-termini of eIF4A3 and Magoh, respectively (Fig. 1c). Co-transfection of eIF4A3-C-SmBiT and MAGOH-C-LgBiT fusions generated an elevated luciferase activity (Fig. 1e, lane 4).

## Mutations known to impair EJC formation impair EJC-NanoBiT activity

To validate the reliability of luciferase activity in relation to EJC formation, we tested mutations that had been shown to impair EJC assembly. MAGOH residues K41 to D43 contact two domains of eIF4A3[25]. MAGOH-C-LgBiT wild-type (WT) or the defective MAGOH KND41-43/A mutants were transiently expressed in cells stably expressing wild-type eIF4A3-C-SmBiT. Although both proteins were expressed at similar levels, much lower luciferase activity is detected with the mutant (Fig. 1f). Next, wild-type and eIF4A3-C-SmBiT T163A, T163D[26] defective mutants were transiently expressed in cells stably expressing wild-type MAGOH C-LgBiT. T163 in eIF4A3 is located next to RNA in the EJC structures[24,25]. Hence, the negative charge introduced by the T163D mutation is much more disruptive than the T163A mutation[26]. Indeed, the latter still provides little more than half the activity of wild type (Fig. 1g, lanes 3 and 4). The eIF4A3 DE401/402KR mutation abolishes a salt bridge between eIF4A3 and Y14[24,25,27]. When wild-type and eIF4A3-C-SmBiT DE401/402KR defective mutants were transiently expressed in cells stably expressing wild-type MAGOH C-LgBiT, much lower luciferase activity is detected with mutant than wild type protein although they are expressed at similar levels (Fig. 1g, lane 7).

These observations demonstrate that the luciferase activity observed when MAGOH-C-LgBiT and eIF4A3-C-SmBiT are co-expressed, reflects the formation of EJC NanoBiT complexes. Therefore stable cell lines co-expressing both wild-type MAGOH-C-LgBiT and eIF4A3-C-SmBiT were established from HEK293 cells.

## EJC particles are very large and contain ribosomes

To further characterize the EJC-NanoBiT particles, we fractionated low ionic strength extracts from stable eIF4A3-C-SmBiT/MAGOH-C-LgBiT expressing cells by ultracentrifugation on sucrose gradients used to fractionate polysomes. Little luciferase activity is detected in the top and bottom fractions (Fig. 2a). It is broadly distributed in most fractions, peaking around the expected size for disomes (D). To exclude a possible bias due to NanoBiT tagging on EJC properties, cell lysates from the parental HEK293 cells were fractionated on sucrose gradients, this time 12 × 1 ml fractions were collected instead of 48 × 0.25 ml ones. Most of eIF4A3, MAGOH and Y14 are found in the top of the gradients (Fig. 2b). Yet, there is clearly a faint peak in the middle of the gradient (fractions 5–8) heavier than 80 S. Importantly, MAGOH and Y14 are efficiently co-immunoprecipitated with an affinity-purified eIF4A3 antibody only from fractions equivalent to those that show the highest levels of luciferase activity (Fig. 2b). We therefore conclude that NanoBiT tagged and genuine EJC complexes display the same distribution upon ultracentrifugation in a sucrose gradient. Noteworthy, a major fraction of EJC core proteins remains unassembled at the top of the gradients.

The average EJC particle size is very large (around 7–8000 kDa). Considering that an average mRNA size around 1400 nt[28], the average mass of the EJC containing mRNP corresponds to approximately 500 kDa per 100 nucleotides, which is very large. In this perspective, it is worth mentioning that the estimated mass of an EJC complex is 350 kDa[29] and that of a nucleosome is 205 kDa for 150 base pairs of DNA whereas the mass of a ribosome is 4300 kDa. We therefore hypothesized that ribosomes account for the large size of EJC-particles. RNA was isolated from immunoprecipitates of pooled gradient fractions containing EJC (Fig. 2b, fractions 5 to 8). Because

commercial HA-antibodies are highly reliable, we used HeLa cells expressing HA-tagged eIF4A3 which were generated through homozygous CRISPR insertion of the HA-tag[30]. Ribosomal RNAs 18 S and 28 S are very efficiently immunoprecipitated when anti-HA antibodies are added to the gradient fractions (Fig. 2c). We conclude that a large proportion EJC-associated mRNP present in cytosolic extracts contain 1 or more ribosomes that contribute to their very large size.

## Ribosomes associated with EJC are stalled

Since translating ribosomes remove EJCs, we treated cells with inhibitors of translation prior to lysis[31]. Cycloheximide freezes elongating ribosomes and polysomes remain unaffected whereas harringtonine stalls translating ribosomes at initiation codons resulting in a strong increase in 40 S, 60 S and 80 S RNA (Fig. 2d, e, dotted lines). Following both treatments, the average EJC particle size is significantly shifted towards higher sizes (Fig. 2d, e, solid lines). Shoulders (stars) that might correspond to monosomes or disomes are reproducibly observed. The increase in average particle size might be interpreted as the additional loading of one ribosome to EJC-bound mRNPs. Harringtonine does not inhibit translation elongation and termination but stalls ribosomes at initiation codons resulting in 80 S mRNP accumulation on translating mRNAs. The EJC ribosome particle size remains higher than 80 S on average, which indicates that its ribosomes are stalled. Noteworthy, a significant proportion of luciferase activity corresponds to EJC-NanoBiT complexes smaller than 80 S and therefore do not contain assembled ribosomes (Fig. 2a). This fraction decreases upon treatment with harringtonine or cycloheximide that increases the amount of 40 S preribosome and 80 S loaded on transcripts (Fig. 2d, e). Taken together, these observations indicate that stalled ribosomes contribute to the very high size of EJC particles present in cytosolic extracts and additional ones are loaded in the presence of translation inhibitors (Fig. 2f).

## Increasing ionic strength relieves constraints in EJC-ribosome particles

Addition of 600 mM NaCl to the lysates slightly shifts the luciferase peak towards smaller particles at the top of gradients (Fig. 3a). But as a whole, large EJC particles resist high ionic strength. Are EJC and ribosomes linked by an RNA molecule? When NanoBiT EJC lysates are digested with a mixture of RNases A and T1, a luciferase peak appears at the top of the gradient (Fig. 3b). But a large amount of NanoBiT EJC particles remains larger than 80 S. This profile remains identical when using three times higher concentrations of RNases disproving a kinetic effect to account for the incomplete transformation of the large complexes into small ones. But when lysates are supplemented with both 600 mM NaCl and RNase, the large NanoBiT EJC complexes disappear completely and are replaced by a peak at the top of the gradients (Fig. 3b). When ionic strength is increased, RNA becomes susceptible to RNases. Note that 80 S ribosomes resist RNase treatments at high salt. The RNase effect strongly suggests that transcripts link EJCs to ribosomes.

While performing these experiments, we noticed that salt addition and RNase treatment of cell lysates resulted in higher luciferase activity. Indeed, it increases 2.5-fold upon RNase digestion (Fig. 3c). The increase is the same when 3 times more RNase is added to the lysate, indicating that a plateau has been reached. Independently, the luciferase activity in the lysate increases up to 3 fold when adding 600–800 mM NaCl. The enhancement is stronger, 9-fold, when RNase is added together with 600–800 mM NaCl. Thus, increasing ionic strength that weakens intermolecular interactions, strongly favours RNase digestion of EJCs and enhances EJC-NanoBiT luciferase activity.

To demonstrate that luciferase fragment fusion is not responsible for the above observations, parental HEK293 cell lysates were fractionated on a gradient. EJC core subunits (eIF4A3, Y14 and MAGOH) are found in the top and in the middle gradient fractions (Fig. 3d, top left).

 

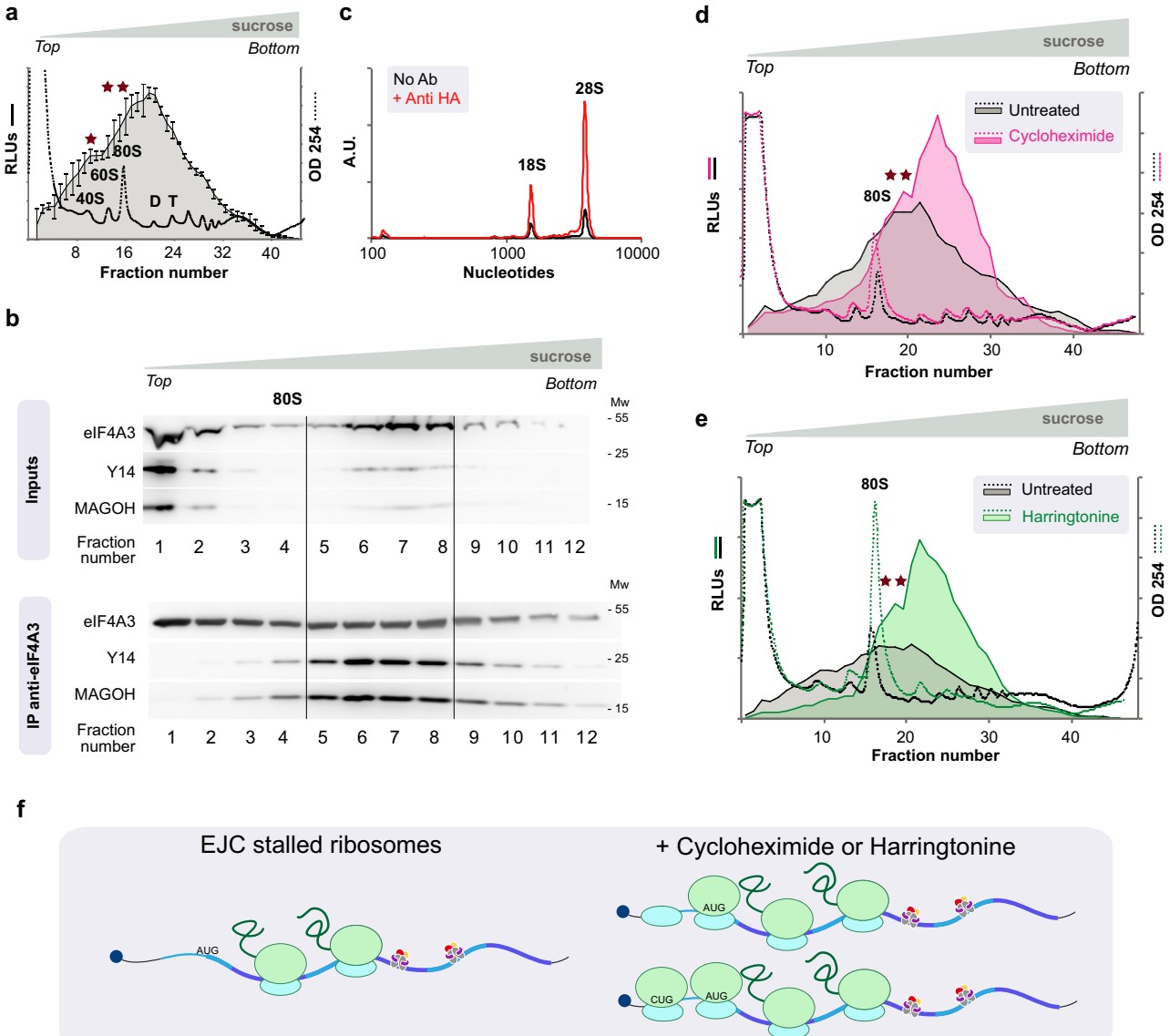

**Fig. 2 | Fractionation of Exon Junction Complex on sucrose gradients. a** Lysates from cells coexpressing MAGOH-LgBiT and eIF4A3-SmBiT (OB9 cells) are fractionated on sucrose gradients. Fractions (48 × 0.25 ml) from HEK 293 EJC-NanoBiT expressing cells were collected after a sucrose gradient (black curve - mean of 3 biological replicates normalized by their total luciferase activity). Mean values +/- standard deviation are plot. **b** Fractions from genuine HEK 293 cell lysates (12 × 1 ml fractions) were collected and analysed by Western blot. Fractions 5 to 9 in 2b correspond to fractions 16 to 36 in 2a, 2d and 2e. eIF4A3, Y14 and MAGOH are detected by Western blot in input fractions and in eIF4A3 immunoprecipitates. **c** Fragment Analyzer profiles of RNAs isolated from precipitates with (red curve) or without (black curve) anti-HA antibodies from pooled 3–12 fractions (12 fractions) of ultracentrifuged HA-eIF4A3 cell lysates. eIF4A3, MAGOH and Y14 proteins are detected on Western blots with specific antibodies diluted 1/1000.

**d** Ultracentrifugation profiles of EJC-NanoBiT expressing cells treated (pink) or not (black) with cycloheximide (100 µg/ml) for 60 min. **e** Ultracentrifugation profiles of EJC-NanoBiT expressing cells treated (green) or not (black) with harringtonine (2 µg/ml) for 60 min. All lysates are fractionated on 11–54% sucrose gradients. RNA in 2b is detected by SYBR green fluorescence. RNA content in 2a, 2d and 2e in 48 × 0.25 ml fractions, is monitored by optical densities at 254 nm (dotted curves) while NanoBiT luciferase activities are shown as continuous lines. Positions of 40 S, 60 S, 80 S disomes (D) and trisomes (T) are indicated. **f** The increased EJC particle size upon treatment with translation inhibitors is attributed to the additional loading of ribosomes arrested at weak non canonical initiation codon or frozen just after initiation and preribosomes queuing upstream. Source data are provided in the Source Data file.

Meanwhile, co-immunoprecipitation is only seen from the middle fractions, which is consistent with the EJC-NanoBiT luciferase activity profile above described (Fig. 3d, bottom left). However, when RNase is added to cell lysates prior to centrifugation, a significant co-immunoprecipitation is obtained from the top fractions although a larger amount remains in the middle fractions (Fig. 3d, bottom middle). Increasing ionic strength alone (600 mM NaCl) hardly affects the gradient profile. However, addition of RNase at high ionic strength, almost suppressed EJC subunits co-immunoprecipitation from middle fractions while strongly increasing it from top of the gradient, peaking

in fraction 2 (Fig. 3d, bottom right). The core EJC remains assembled in these harsh conditions.

In summary, EJC ribosome particles resist RNase digestion at low ionic strength. These particles appear to be constrained and an increase in ionic strength alleviates the constraints.

## EJC disassembly occurs through a fast translation-dependent pathway and a slow translation-independent pathway
We next investigated the dynamics of EJC disassembly. NanoBiT activities were measured directly adding its substrate to crude cell lysates.

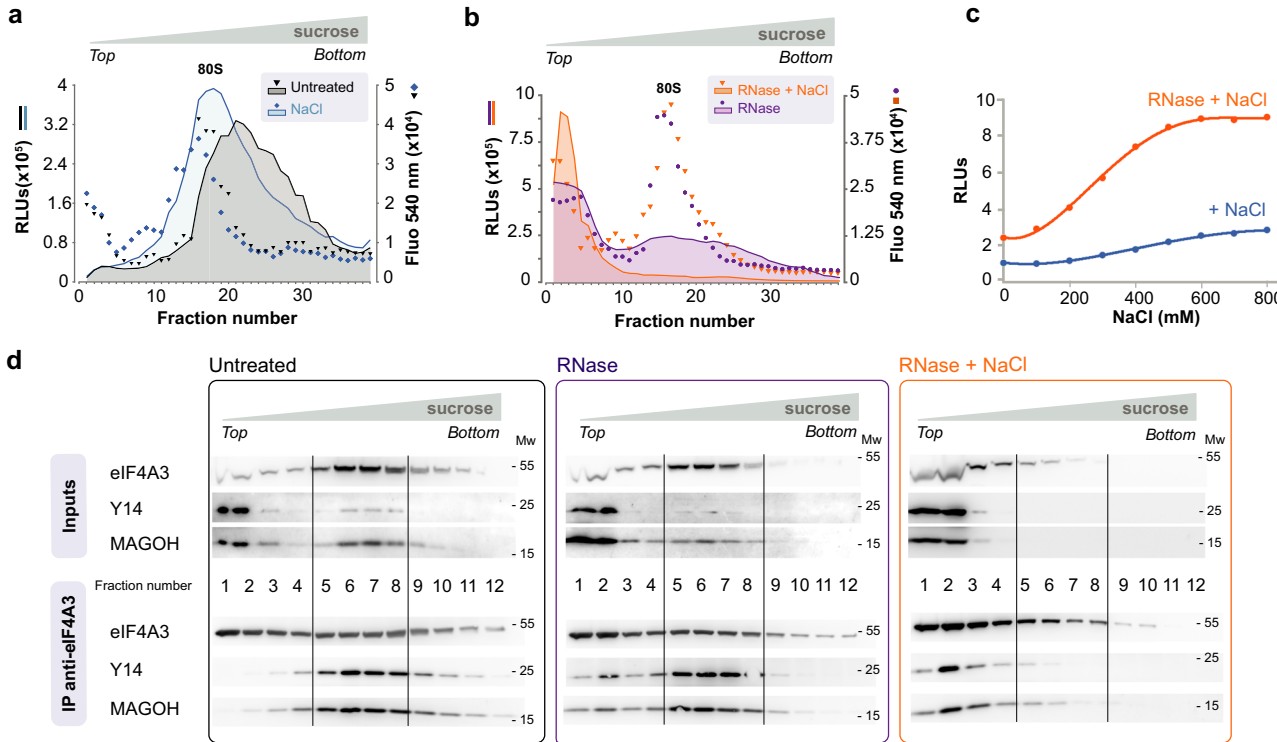

**Fig. 3 | RNase and ionic strength susceptibilities of Exon Junction Complexes.**
**a**, **b** Lysates from cells coexpressing MAGOH-LgBiT and eIF4A3-SmBiT (OB9 cells) are fractionated on sucrose gradients. Luciferase activity (RLUs) and SYBR green fluorescence at 540 nm are measured in gradient fractions. **a** Control lysates (black), lysates with NaCl (600 mM) (light blue), **b** RNase (purple) or both RNase and NaCl (orange) are added to lysates from EJC-NanoBiT expressing cells. **c** Luciferase activity in crude lysates supplemented with 600 mM NaCl (blue) or NaCl and RNase (orange). RNA in 3a, 3b and 3c are detected by SYBR green fluorescence emission at 540 nm (dots) and NanoBiT luciferase activities (continuous lines) are measured in 48 × 0.25 ml gradient fractions. **d** Lysates from HEK293 cells are fractionated on 11-54 % sucrose gradients. EJC core proteins are detected by Western blot in input fractions (top) or co-immunoprecipitated from the same fractions by anti-eIF4A3 (below). eIF4A3, MAGOH and Y14 proteins are detected on Western blots with specific antibodies diluted 1/1000. All these experiments have been repeated twice with similar results. Source data are provided in the Source Data file.

The luciferase activity remains constant along a few hours of cell culture corresponding to a steady state when EJC assembly and disassembly rates are equal. Data are normalized for each experiment as the ratio of luciferase activity at a given time point (EJC) divided by the steady state activity in untreated cell lysate ($EJC^O$). When transcription is arrested, no new transcripts are made and subsequently spliced. Hence, no new EJCs is expected to be assembled and deposited at exon-exon junctions. A decrease in EJC levels due to its disassembly is expected. Indeed, when cells are lysed after adding an inhibitor of transcription such as 5,6-Dichloro-l-β-ribofuranosylbenzimidazole (DRB) or actinomycin D (ActD), to the culture medium, the luciferase activity in lysates decreases with duration of the treatment (Fig. 4a). The decrease follows the same kinetic within experimental errors with both inhibitors. Transcription is the common denominator as these compounds act through distinct mechanisms: actinomycin D intercalates into DNA, whereas DRB inhibits the positive transcription elongation factor (P-TEFb), a protein kinase required to elongate transcription and to couple transcript maturation with transcription[32]. Experimental data up to 150 min is nicely fit (solid curve - Fig. 4a) by a sum of two exponentials, a fast and a slow one. Amplitude and time constant of the slow exponential are determined using time points from 60 to 150 min (dotted curve). Characteristics of the fast exponential are determined from the difference between experimental data and computed slow exponential contribution from 15 to 45 min.

If translation contributes to remove EJCs, translation inhibition should slow down their disassembly. When actinomycin D (or DRB) and cycloheximide are added simultaneously, the decrease in EJC-NanoBiT content is nicely fit by a single exponential with a time constant of 176 ± 10 min, which is consistent with that of the "slow" exponential described above (Fig. 4b). Hence "slow" EJC disassembly does not require removal by translating ribosomes. In contrast, the "fast" initial EJC-NanoBiT decrease is not observed anymore and therefore likely requires translating ribosomes.

Pooling actinomycin D and DRB data, we evaluate average time constants of 12 ± 2 min for the "fast" process and 200 ± 40 min for the "slow" one (Fig. 4c). Errors are estimated from determinations of exponential parameters for the actinomycin and DRB data sets separately. "Fast" EJC disassembly is 17 fold faster than the slow one. The sum of fast and slow exponentials is set equal to 100 (%) 5 min after addition of the drugs. This takes into account a lag most likely reflecting that drug action is not immediate. The amplitude of the "slow" decrease at 5 min is 69 ± 2%. The "slow" disassembly pathway concerns 2/3rd of the EJCs. Translation-dependent EJC disassembly is more than 15-fold faster than the translation-independent one and concerns only 31 ± 2 % of the EJCs present at steady state.

## Assembly of EJCs prone to translation-dependent disassembly is also a fast processes

We next questionned the probability of a newly deposited EJC to be disassembled by a translation-dependent (TD) or a translation-independent (TinD) mechanism. The respective contributions of these pathways may be estimated assuming that the disassembly rate is proportional to EJC concentration ($R_{diss} = k_{diss} (EJC)$). At steady state the net reaction rate is zero, the EJC assembly rate ($R_{ass}$) is equal to its disassembly rate ($R_{diss}$), the EJC concentration ($EJC^O$) is constant over time. Assuming that one distinguishes two EJC populations, those that are disassembled slowly independently of translation ($EJC^{TinD}$) and those that are disassembled rapidly and depending on translation ($EJC^{TD}$).

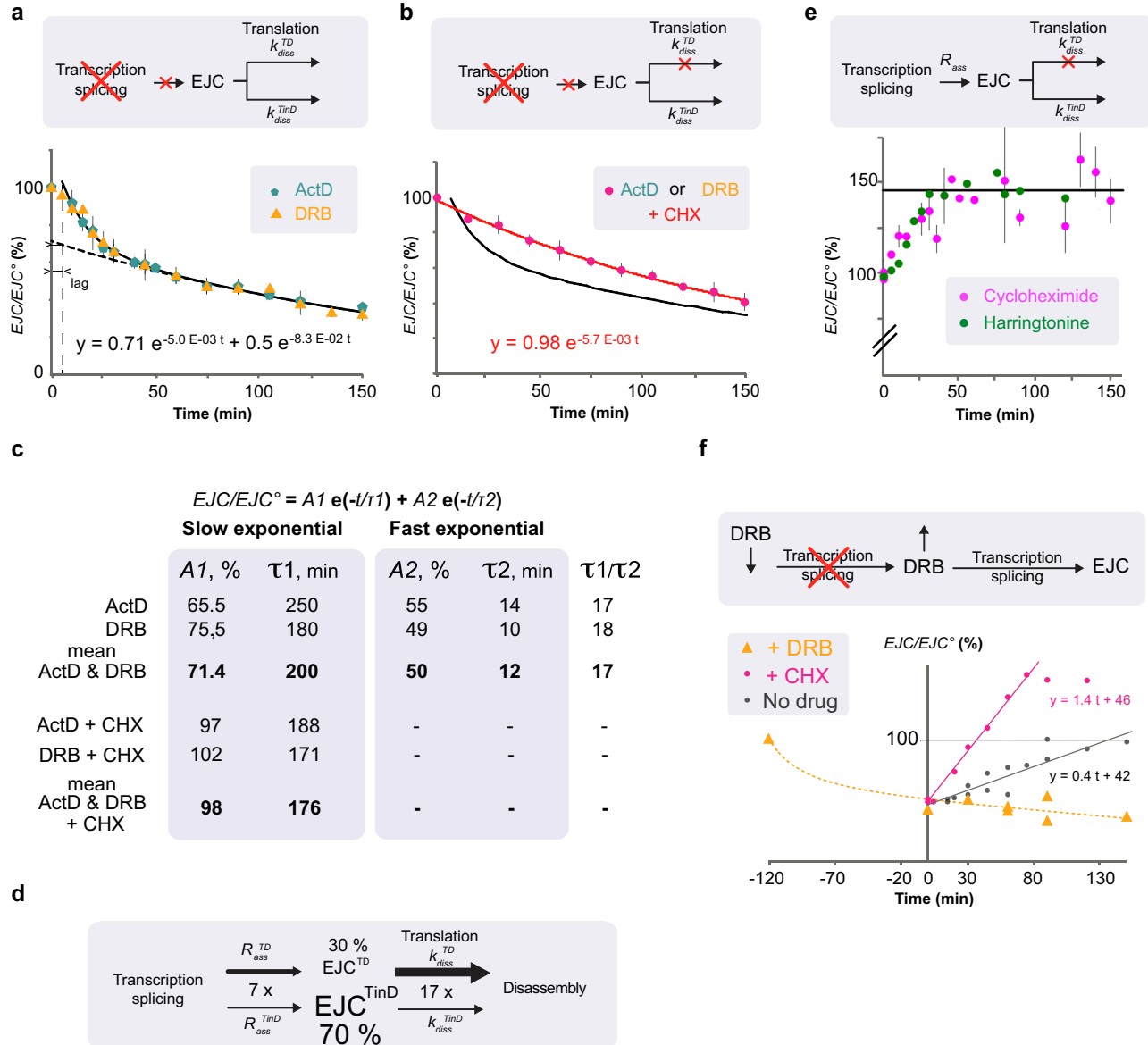

**Fig. 4 | Kinetics of Exon Junction Complexes assembly and disassembly.** Time-courses for luciferase activities in lysates from cells stably expressing both eIF4A3-SmBiT and MAGOH-LgBiT (OB9 cells) exposed to **a** inhibitors of transcription; 5,6-dichloro-l-β-ribofuranosylbenzimidazole (DRB) and actinomycin D (ActD). The black doted curve corresponds to the slow exponential decrease. The black continuous curve corresponds to the best fit of experimental points by two exponentials; **b** cells exposed simultaneously to inhibitors of transcription (ActD or DRB) and translation (CHX) (pink dots); the continuous black curve corresponds to transcription inhibition as in B; **c** Amplitudes (A) and time constants (τ) of the exponentials describing the decrease in EJC-NanoBiT activity following an arrest in transcription and/or translation. Amplitudes (A1 or A2) and time constants in minutes (τ1 or τ2) correspond to the best fits obtained using indicated data sets shown in **a**, **b**. **d** Scheme illustrating why translation-dependent disassembly accounts for only 30% EJC at steady state yet 85% of newly deposited EJCs are

disassembled by such pathway. Assembly rates ($R_{ass}$) and disassembly ($k_{diss}$) rate constants for EJCs prone to fast translation dependent (TD) and slow translation independent disassembly (TinD), respectively. As shown in panel **c**, $k^{TinD}_{diss} = 1/\tau1$ and $k^{TD}_{diss} = 1/\tau2$ and $k^{TD}_{diss} = 17 \times k^{TinD}_{diss}$. Therefore $R^{TD}_{ass} = 7 \times R^{TinD}_{ass}$. **e** Increase in luciferase activity in lysates from cells exposed to inhibitors of translation; cyclo-heximide (CHX) or harringtonine **f** Recovery of luciferase activity when DRB containing media are replaced after 2 h of incubation by fresh media with (pink circles) or without (black dots) cycloheximide (CHX). Yellow triangles correspond to cells treated with DRB until lysis. The yellow dotted curve corresponds to the best fit determined in **b**. Each point corresponds to technical replicates normalized by NanoBiT activity of untreated cells from the same 12-well plate. Initial points of recovery are fit by straight lines. Mean values±standard deviation are plot in **a**, **b**, **e**. Source data are provided in the Source Data file.

At steady state,

$$R_{ass}^{TinD} = R_{diss}^{TinD} = k_{diss}^{TinD}(EJC^{O-TinD}) \text{ and } R_{ass}^{TD} = R_{diss}^{TD} = k_{diss}^{TD}(EJC^{O-TD}) \quad (1)$$

$$R_{ass}^{TD}/R_{ass}^{TinD} = R_{diss}^{TD}/R_{diss}^{TinD}$$
$$= (k_{diss}^{TD}/k_{diss}^{TinD})(EJC^{O-TD})/(EJC^{O-TinD}) \quad (2)$$

According to the data provided in Fig. 4c $(EJC^{O-TD})/(EJC^{O-TinD}) \approx 30/70$ and

$$(k_{diss}^{TD}/k_{diss}^{TinD}) = \tau1/\tau2 \approx 17 \quad (3)$$

leading to:

$$R_{ass}^{TD}/R_{ass}^{TinD} \approx 17 \times (30/70) \approx 7 \quad (4)$$

The assembly rate for EJCs that undergo fast translation-dependent (TD) disassembly is 7-fold faster than that of those that undergo slow translation-independent (TinD) disassembly. Steady state concentrations are the result of opposed assembly and disassembly processes. Although their assembly is 7-fold faster, "unstable" EJCs accumulate less than "stable" ones because their dissociation is much faster (17-fold) than that of "unstable" ones (Fig. 4d). Paradoxically at steady state most, 7 out 8 newly deposited EJCs, are likely disassembled by a fast translation-dependent process. EJC removal from mRNA had already been proposed to follow both translation-dependent and translation-independent mechanisms[19]. We now provide an estimate of their respective contributions.

### Observing assembly of the EJC following resumption of transcription

As translating ribosomes are thought to remove the EJC[17,19], one expects to observe an increase in EJC levels when protein synthesis is inhibited. When cells are lysed after addition of a translation inhibitor to the culture medium, the luciferase activity in the lysate increases up to $140 \pm 5$ % (horizontal line) but reaches a plateau within 30 min of treatment (Fig. 4e). Identical observations are obtained using either harringtonine or cycloheximide, acting through different mechanisms. The value of the plateau is much weaker than expected from the above estimates for assembly rates. Some component critical for assembly might be depleted or transcription/splicing rates decrease upon inhibition of translation.

Attempting to investigate EJC assembly directly, we followed its formation after removal of transcription inhibition. DRB is reversible and well adapted to transcription recovery studies and has successfully been used to evaluate rates of transcription and splicing in live human cells[33]. Thus, after 2 h of treatment the DRB is washed away and replaced with fresh medium. The luciferase activity then increases back rapidly (Fig. 4f). If DRB-containing medium is replaced by cycloheximide-containing medium, the recovery is 3.5-fold faster globally during the first hour and plateaus. A rigorous mathematical treatment is quite tricky and not justified given experimental uncertainties and poor knowledge in transcription/splicing recovery delays following inhibition of transcription. Furthermore, transcriptional stress is released, specific transcripts might be transcribed and these transcripts might be less susceptible than steady state ones to fast translation-dependent EJC disassembly. Nevertheles we performed a simple numerical analysis (Supplementary methods 1) using the numbers from Fig. 4c to estimate the rates when the recovery has generated 0.75 ($EJC^0$), starting from 0.5 ($EJC^0$). In these conditions, we predict a recovery rate in the presence of cycloheximide 3.4-fold faster than without, which compares well with the experimental data. The recovery is faster in the presence of cycloheximide in consistency with the significant contribution of a translation-dependent disassembly. This experiment shows that new EJCs are assembled as soon as transcription resumes. EJC assembly involves transcription but this process does not require protein neo-synthesis. It illustrates the common view that transcription is a requirement for EJC assembly.

### RIP-sequencing of EJC bound transcripts

The split-luciferase assay provides a global view on EJC behaviour, it does not provide any indication about the behaviour of specific transcripts. To investigate the persistence of EJCs at the transcript level, we performed RIP-sequencing using a cell line where eIF4A3 coding genes have been edited with a N-terminal HA-tag (Fig. 5a). In brief, low salt lysates are fractionated by ultracentrifugation as above. An anti-tag antibody is added (or not) to pooled gradient fractions containing most of the EJCs. RNAs isolated from beads are oligo-dT purified, sequenced and reads are mapped to a reference genome where pseudogenes have been masked[34]. For each gene (g), we determine sequence-depth normalized read numbers (RPM) for immunoprecipitated RNAs ($Ab(g)$) and blank RNAs ($blk(g)$) precipitated by protein A in the absence of antibodies (Supplementary Data 1). For reliability, we filtered out those genes that had less than 50 RPM mapped on average in both replicates for immunoprecipitated RNAs from untreated cells ($Ab0(g) > 50$). 4326 genes satisfy this requirement (Supplementary Data 2). Among these 3/4 (3110) are more abundant in immunoprecipitates than blank precipitates ($Ab0(g) > blk(g)$) (Fig. 5c). Their enrichment ratio ($EnrO(g) = Ab0(g)/blk0(g)$) is higher than 1. Highly expressed transcripts have a tendency to be less enriched than others. Since a high proportion of transcripts are immunoprecipitated, a normalization procedure based on definition of invariant genes (g') is required to evaluate the "true" number of EJC-associated transcripts ($Ejc(g)$)[35].

The most abundant transcripts in the input gradient fractions from untreated cell lysates (Inp0) are those coded by mitochondrial genes (mt-g) (Fig. 5b, Supplementary Fig. 1a and Supplementary Data 2). Enrichment ratios are lower for mitochondrial transcripts (mt-g) than for the majority of nuclear transcripts suggesting that most non-mitochondrial encoded transcripts are enriched by anti-EJC immunoprecipitation (Fig. 5c, d and Supplementary Fig. 1b). Since mitochondrial transcripts are unlikely to interact with nuclear-encoded RNAs, we used them as internal "spike-in" invariants. Thus we estimated normalization factors "α" for every replicate (Supplementary Fig. 1c and Supplementary Data 3) fitting the mitochondrial transcript data as:

$$Ab(mt - g) - \alpha\,blk(mt - g) = 0 \qquad (5)$$

this estimate is used to evaluate the "true" number of EJC-bound transcripts for any gene (g):

$$Ejc(g) = Ab(g) - \alpha\,blk(g) \qquad (6)$$

To begin with, we dealt with transcripts from exponentially growing cell lysates (0 h with DRB). There is a weak correlation between $Ejc0(g)$ read values and input read values (Fig. 5e). There is a good correlation between $Ejc0(g)$ values obtained for two biological replicates performed on different days (Fig. 5f).

### Transcript-specific binding to EJC

$Ab0(g)$, $blk0(g)$ and $Ejc0(g)$ integrate numbers for reads mapped along entire genes. To get a closer insight on individual transcript characteristics, we used the Integrative Genome Viewer (IGV)[36]. We focussed on the 546 genes that on average show more than a 4-fold enrichment ($Ab0(g)/blk0(g) > 4$) and are therefore undisputedly bound to the EJC. The 50 most abundant genes in replicates I and II were systematically inspected using the Integrative Genome Viewer (IGV) comparing immunoprecipitated (Ab0) and blank (blk0) tracks. A few typical examples are shown in Fig. 5g–i. Adjacent genes such as *NOP56* and *IDH3B* with similar expression levels can be seen on the same screenshot and yet illustrate distinct characteristics (Fig. 5g). Although they have similar abundance in inputs, *NOP56* transcripts are efficiently bound by EJC whereas *IDH3B* ones are not. Furthermore in several cases, EJC-bound and input transcripts display distinct characteristics. For instance, the *NOP56* input profile is consistent with a mixture of two major transcripts that have been described[37]. One of them initiates with snoRD86 as the result of nonsense mediated decay processing of its 5' extremity. The latter dominates in EJC immunoprecipitates. *CIRBP* is another example of alternatively spliced transcripts enriched in EJC particles (Fig. 5h). Input reads are consistent with a major representation of some of its 46 RNA species annotated in ENSEMBL. But EJC immunoprecipitates are enriched in a less spliced one (exon in red). Alternatively spliced transcripts such as *CIRBP* or *NOP56* are enriched with the EJC in consistency with previous observations of Kovalak et al.[38]. However, unlike these two examples, most transcripts show

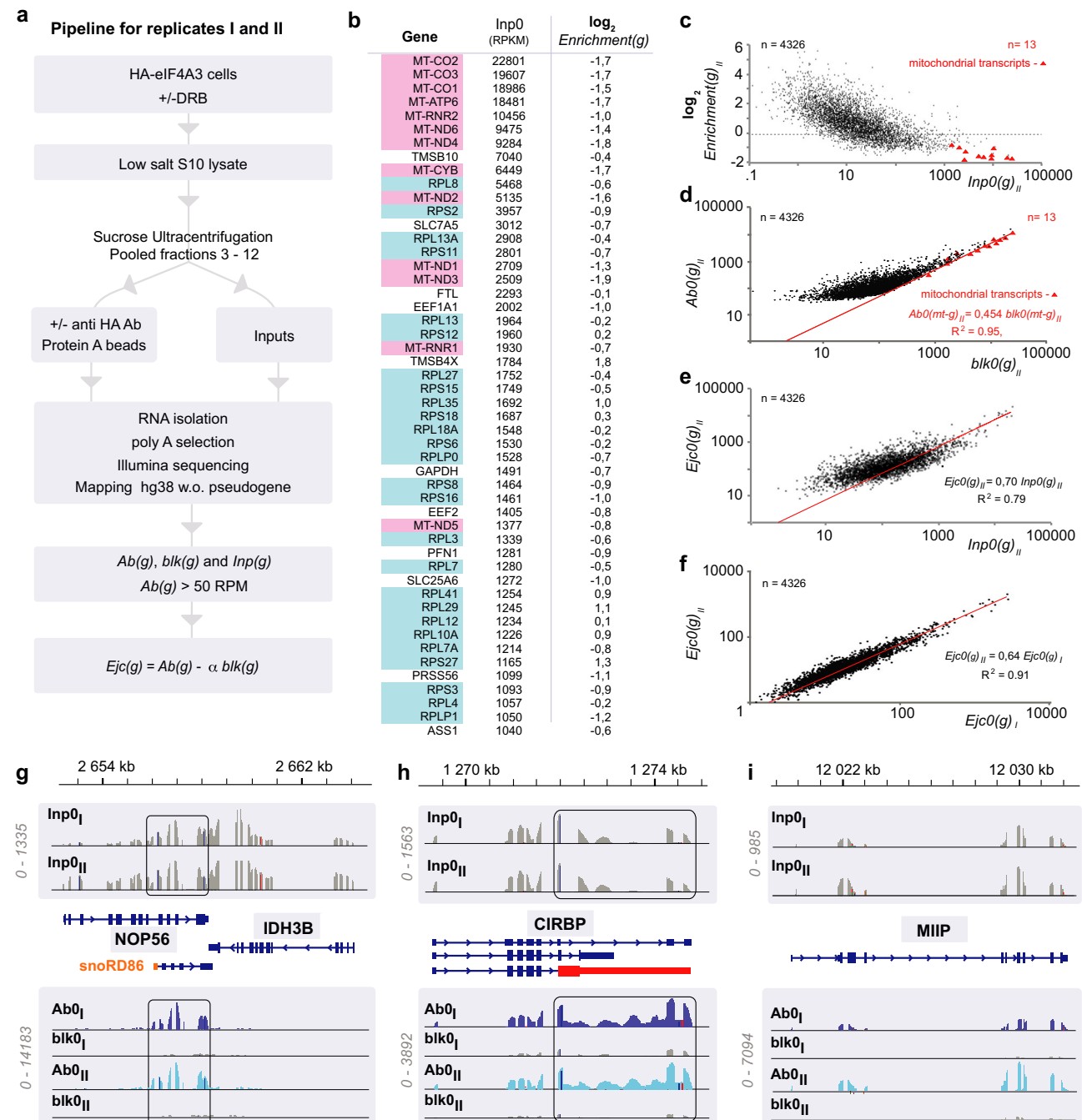

**Fig. 5 | Transcripts immunoprecipitated with EJCs. a** Pipeline for RNA isolation from HA-eIF4A3 cells and sequencing of replicates I and II; **b** The left column lists the 50 most abundant transcripts found in inputs for replicates I and II (data from Supplementary Data file 2) ranked according to decreasing average reads per kilobase per million (RPKM) (middle column), the right column provides the read enrichment upon immunoprecipitation. Mitochondrial transcripts are in mauve. Ribosomal protein coding transcripts are in blue. The middle column indicates the corresponding RPKM values. Right column indicates the corresponding enrichments (Enrichment(g) = $Ab0(g)/blk0(g)$); **c** Enrichments are plotted against input (pooled gradient fractions) reads per gene ($Ab0(g)/blk0(g)$). Transcripts above the dotted line are enriched more than 1; **d** Immunoprecipitated reads ($Ab0(g)_{II}$) are plot versus blank precipitated reads ($blk0(g)_{II}$) for replicate II. Red triangles correspond to mitochondrial-encoded transcripts ($mt$-$g$) (data Supplementary Data file 3). **e** Poor correlation between Input reads per gene ($Inp0(g)_{II}$) and estimated reads bound to EJCs ($Ejc0(g)_{II}$) for replicate II. **f** Correlation between estimated $Ejc(0)_I$ and $Ejc(0)_{II}$ from replicates I and II (data for D and E Supplementary Data file 4). **g–i** IGV profiles for *NOP56, MIIP* and *CIRBP* transcripts for replicates I and II. Reads immunoprecipitated with (Ab0) or without (blk0) antibodies, and inputs (Inp0) are grouped separately. Read scales are shown on the left of each profiles. *snoRD86* in *NOP56* transcript is labelled in orange. The deficient CIRBP splicing is labelled in red. Source data are provided in the Source Data file.

similar input and EJC-bound splicing profiles like *MIIP* (Fig. 5i). To increase confidence in these observations, we analysed the same genes by IGV for replicates III and IV, which are prepared following a distinct pipeline (Supplementary Fig. 1f). In brief, RNAs are immunoprecipitated using anti-HA-coated beads from fractionated HA-tagged eIF4A3 cell lysates, RNAs immunoprecipitated from parental HeLa cell lysates are used as negative controls. Observations obtained with replicates III and IV are highly similar to those with replicates I and II (Supplementary Fig. 1g–i). Thus the transcript-specific extent and profile of EJC binding is highly reproducible.

### EJC-bound transcripts linked to microtubules and centrosomes

Transcripts for housekeeping genes such as ribosomal protein coding, eukaryotic elongation factors, *ACTB* and *GAPDH* make up the most abundant ones in the inputs (Fig. 5b, Supplementary Data 4). Most are poorly enriched with the EJC as $\log_2(Enrichment(g) < 0$. Nevertheless, their enrichment remains significant ($>-1$), it is higher than enrichment of mitochondrial transcripts ($<-1$) that do not undergo nuclear splicing hence EJC deposition. The high abundance in housekeeping gene transcripts in the inputs leads to a high background (*blk*) immuno-precipitation hence low enrichment. Overall, there is poor correlation between transcript abundance in inputs ($InpO(g)$) and bound to EJC ($Ejc^0(g)$) (Fig. 5e). We could not detect any straightforward features related to transcript binding to the EJC: No significant links to exon number, gene length, transcript length, or stability. Several non-coding RNAs such as *SNHG* (1, 3, 4, 12, 19) and *LINC00173* are enriched more than 4-fold (Supplementary Data 4). Transcripts that have not been spliced are not expected to associate an EJC. Indeed, out of the 18 annotated intron-less transcripts found among the 4326 input ones (with $AbO(g)>50$ RPM on average) only one, *HEXIM1*, is significantly immunoprecipitated ($AbO(g)/blkO(g) > 1$). However several *HEXIM1* reads enriched with EJC are mapped to transcripts that have undergone unreported splicing events (Supplementary Fig. 1e).

According to a gene ontology analysis of the genes enriched with EJCs. the most enriched ones correspond to the methyl transferase complex, cytoplasmic stress granules, spliceosomal complex nuclear speckles, centrioles, microtubules organizing center and centrosomes (Supplementary Data 5)[39]. Proteins encoded by 72 out of 546 EJC-associated transcripts enriched on average more than 4-fold are associated with centrosomes, microtubules and/or ciliogenesis (Supplementary Table 1). The functional category of transcripts strongly enriched with EJCs is clearly biased.

### Transcript-specific persistence with EJCs

To evaluate the persistence of individual transcripts bound to an EJC following an arrest in transcription, we determined $Ejc^0(g)$, $Ejc^{60}(g)$ and $Ejc^{120}(g)$ following no treatment (0), 1 h (60) or 2 h (120) exposure to DRB for both replicates I and II, respectively. The decrease in $Ejc(g)$ following an arrest in transcription hence EJC assembly, reflects the persistence of transcript (g) with the EJC. Again, we focussed on the 546 genes that show at least a 4 fold average enrichment ($AbO(g)/blkO(g) > 4$) as their $Ejc(g)$ determinations are more reliable since they are less dependent upon blank correction. On average, $Ejc^0(g)$ is expected to be larger than $Ejc^{60}(g)$, itself larger than $Ejc^{120}(g)$ since no new EJCs assemble when transcription is arrested (Fig. 6a). The ratios $Ejc^{120}(g)/Ejc^0(g)$ and $Ejc^{60}(g)/Ejc^0(g)$ provide estimates for the persistence of a transcript "g" with EJCs (Supplementary Data 6). These ratios decrease by one order of magnitude for some transcripts compared to others. For some genes, it exceeds 1. Such an apparent increase after transcription arrest is likely due a spike-in normalization bias. RNA stability can be evaluated from reads in inputs using the ratio $Inp^{120}(g)/Inp^0(g)$. RNA stability and persistence with EJC are not correlated (Fig. 6b). When the mean $Ejc^{60}(g)/Ejc^0(g)$ ratio is plotted versus $Ejc^{120}(g)/Ejc^0(g)$, a good correlation between the two sets is observed (Fig. 6a). Overall, genes that are most affected by a transcriptional arrest of 2 h are the most affected by 1 h of arrest.

To get a closer insight, we looked at the behaviour of typical individual transcripts using Integrative Genome Viewer (IGV). In Fig. 6c–h, we show data for 6 individual transcripts illustrating the distinct behaviours observed in Supplementary Data 6. Two extreme classes of EJC-bound transcripts are distinguishable: short-lived ones such as *PNRC2* that strongly decrease in inputs as well as immuno-precipitates upon DRB treatment (Fig. 6c) and persistent ones such as *MIIP* that do not decrease in either inputs or immunoprecipitates following an arrest in transcription (Fig. 6d). Some transcripts such as *SNHG19* are non coding, strongly decrease in immunoprecipitates but

remain stable in inputs (Fig. 6e). *SNHG19* and *RAB26* are close neighbouring genes immunoprecipitated with similar efficiencies. In contrast to *RAB26* transcripts, *SNHG19* transcripts bound to EJC decrease markedly in response to an arrest in transcription. EJC enrichment in alternative *CIRBP* transcripts (red stars) decreases faster after DRB than that of other transcripts thereby illustrating a differential persistence time within transcripts coded by the same gene (Fig. 6f). Note that alternative transcripts are faintly seen in untreated cell inputs and disappear from DRB-treated cell inputs, thereby indicating a faster degradation than that of the other ones. In contrast, *SRSF7* provides an example of alternative transcript enriched with EJC but all transcripts decrease at a similar rate (Fig. 6g). Furthermore, *NOP56* is an extreme case where alternative transcript/EJC association is very stable (Fig. 6h). Similar observations are made with two additional data sets (replicates III and IV) performed using anti HA tag beads illustrating the robustness of our observations (Supplementary Fig. 2a–f).

Four non-coding small nucleolar RNA host gene transcripts (SNHGs') are among the 18 (out of 546) less persistent EJC-associated transcripts (Supplementary Data 6). In contrast, 24 out of the 100 more persistent EJC-associated transcripts code for proteins linked to centrosomes, microtubules and/or ciliogenesis (Supplementary Table 1 and Supplementary Data 5). Taken together these observations demonstrate that the lifetime of an EJC-RNA complex is transcript-dependent and that there is a transcript ontology bias in strongly enriched EJC bound transcripts.

## Discussion

Split-nanoluciferase (NanoBiT) is a powerful tool to investigate EJC properties and dynamics. When two complementary fragments of nanoluciferase are fused to two subunits (eIF4A3 and MAGOH) of the EJC core complex, a strong luciferase activity is recapitulated, indicating that both fusion proteins belong to the same complex. The reliability of luciferase activity reflecting non-tagged endogenous EJC is supported by several arguments. First, known assembly deficient mutations of either subunit kill this activity. Second, the distribution and size of the EJC-NanoBiT complexes in a sucrose gradient is the same as that of endogenous EJC defined by co-immunoprecipitation. Third, RNase treatment and/or ionic strength have the same effects on the sizes of light-emitting complexes and immunoprecipitated endogenous EJC. Sensitivity, easiness and quantitation are major improvements of the EJC-NanoBiT system over previously available immunoprecipitation methods. Split-luciferase assay have mainly been used to screen for inhibitors. This study illustrates the possibility to investigate the properties and dynamics of RNA protein complexes.

EJCs are stably associated with stalled ribosomes. EJC complexes from cytosolic extracts are found broadly distributed peaking around 7000 kDa (a little less than disomes). This is in consistency with previously published experiments showing that EJC complexes elute in the void volume of a Sephacryl column ($> 2000$ kDa)[40,41]. In these studies, the large size of EJC complexes had been attributed to large stoichiometric excess of RNA binding proteins such as SR proteins that might coat EJC-bound transcripts. However, the size distribution of mRNAs in mammalian cells peaks around 1400 nt[28]. As EJC particles containing more than one mRNA molecule had not been detected in previous studies, one expects 500 kDa of associated components to be present on average per 100 nt of transcript. This is huge for proteins that are 50 kDa on average. Indeed, mRNPs have been shown to contain only 50-80% proteins by weight which corresponds to 20 kDa per 100 nt[42]. EJCs co-precipitate with high amounts of ribosomal RNAs (this work) as well as ribosomal proteins[43]. The presence of ribosomes (4300 kDa per ribosome) with EJC particles accounts for their large size. However, a small amount EJC particles devoid of ribosomes are found in low salt cytosolic extracts (this work) and 2000 kDa EJCs associated with the transcription export complex have been isolated from nuclear extracts[44].

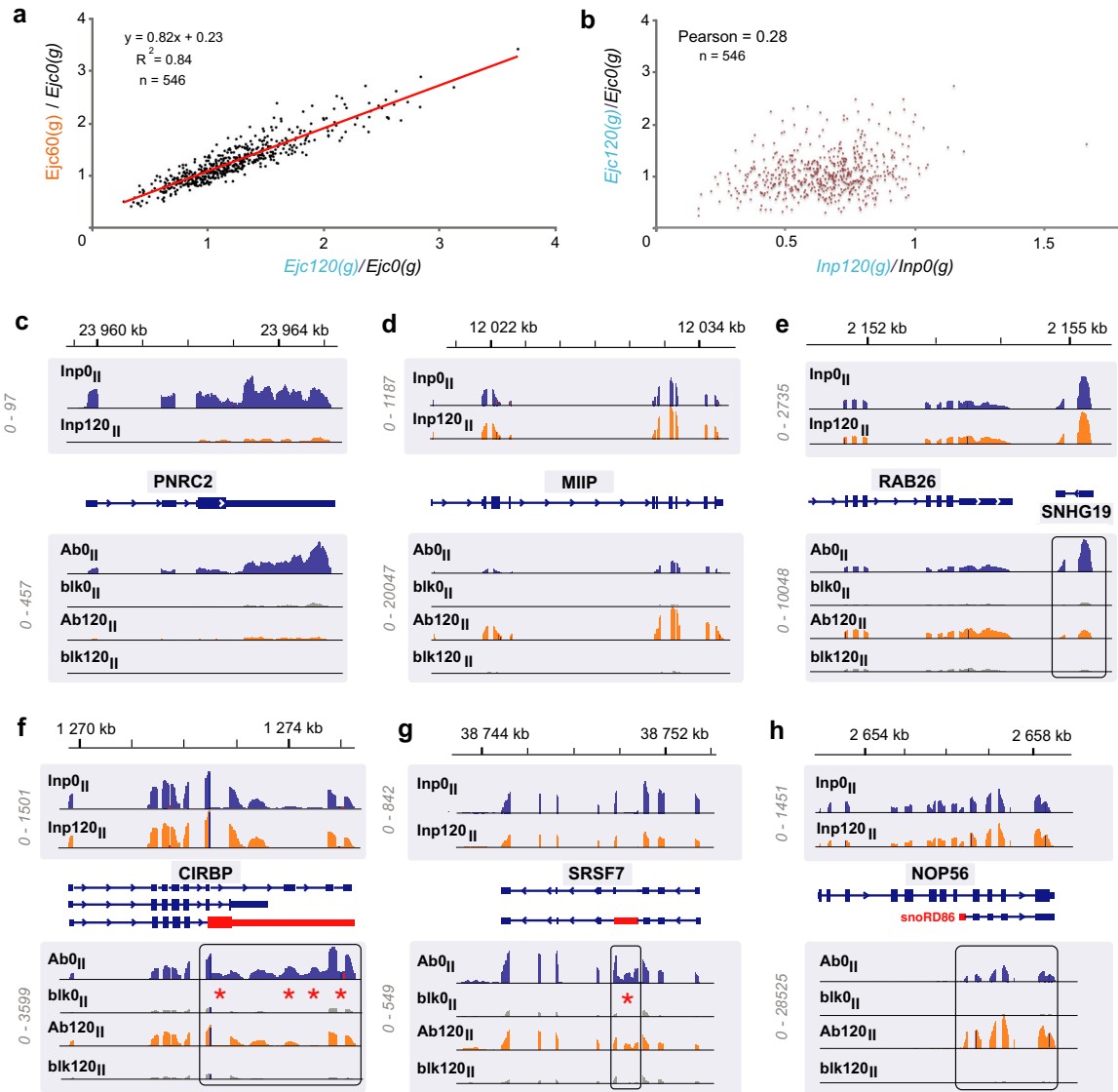

**Fig. 6 | Persistence of EJC binding is transcript specific. a** Correlation between persistence indexes *(Ejc120/Ejc0)* and *(Ejc60/Ejc0)* averaged for replicates I and II. For **a**, **b**, only the 546 transcripts showing an enrichment *Ab0(g)/blk0(g) > 4* are taken into account. *Ejc0(g)* correspond to untreated cells, Ejc60(g) after 1h00 and Ejc120(g) after 2h00 of DRB treatment. Data from Supplementary Data file 6. **b** Poor correlation between persistence indexes *(Ejc120/Ejc0)* averaged for replicates I and II and inputs averaged for the same replicates. **c–h** IGV immunoprecipitated and input profiles without DRB (blue) or following 2h00 DRB treatment (orange) are shown for typical transcripts of replicates I and II. Red stars mark *CIRBP* and *SRSF7* reads specifically enriched in immunoprecipitates. Blank precipitated reads are in grey. Read scales are shown on the left of each profiles. Source data are provided in the Source Data file.

In the presence of harringtonine, elongating ribosomes are expected to terminate normally and drop off. Reinitiating ribosomes arrest at the start codon as 80 S particles. Therefore an EJC particle size remaining well above 80 S, demonstrates that its ribosomes do not elongate translation and drop off. They are stalled. On the contrary, an increased EJC particle size is observed upon inhibition of translation. Both cycloheximide and harringtonine treatments can lead to queuing of preinitiation complexes and enhance recognition of weak non-AUG start codons[45]. Hence the increased EJC particle size might be attributed to the additional loading of ribosomes arrested at weak non-canonical initiation codon (harringtonine) or frozen just after initiation (cycloheximide) and pre-ribosomes queuing upstream (Fig. 2f).

Many transcripts remain stably associated with EJC-ribosome particles up to several hours. Such finding was totally unexpected as EJCs are commonly associated with non-sense mediated RNA decay (NMD). Translating ribosomes are supposed to remove EJCs and NMD

occurs when an EJC remains as it positioned after a stop codon[3]. NMD requires termination of translation but the EJC linked ribosomes described in this study are likely stalled without termination factors. It remains an open question how EJC-ribosome bound transcripts escape the no-go decay (NGD) process triggered by ribosome stalling[46].

EJCs are constrained. The size of EJC-ribosome particles is little affected by RNase treatment in low salt containing buffer. It has been previously reported that large EJCs complexes resist endonuclease digestion[40,41]. RNA fragments of 30-150 nt are protected from RNAse digestion in large EJC complexes ( > 2000 kDa) whereas those protected in "free" EJC complexes (450 kDa) are much smaller. Ribosomes are very large structures that shield 28 to 30 nt from nuclease digestion[47]. Compaction of the EJC-ribosome particles might limit access to the nucleases between ribosomes and EJC and also restrict the EJC-NanoBiT activity. Increasing ionic strength loosens protein/protein and protein/RNA interactions leading to a partial release of the

constraints likely responsible for the increased accessibility to RNase. Thus, high ionic strength is required for an RNase treatment to completely detach the EJC from ribosomes. In addition, luciferase activity increases markedly upon RNase digestion at high ionic strength. The split luciferase fragments should not be prevented to interact by neighbouring macromolecules. The EJC-luciferase gives a global view, some EJC split fragments under constraint might not be able to interact at all whereas others may. As ionic strength and RNase digestion weaken protein interactions, release the ribosomes and therefore removes constraints preventing efficient split-luciferase matching. Noteworthy, in the nucleus, EJCs limit access to m6A RNA modification and reside within the interior of an export complex[11,44]. Here we show that constrained EJCs are associated with stalled ribosomes. It is commonly thought that mRNPs initially form compacted structures that are decompacted by translating ribosomes[42]. Stalling likely prevents ribosomes from decompacting the EJC mRNPs.

EJC disassembly follows translation-dependent and independent processes. Removal of the EJC by the first incoming ribosome is a common view but the concurrent contribution of a translation-independent pathway has also been formulated[3]. Thanks to quantitation and sensitivity of the EJC-NanoBiT system, it is now possible to assess the relative contribution of both pathways. When transcription is arrested EJC are not assembled anymore and luciferase activity decreases following two exponential phases. Inhibition of translation only partially prevents the decrease in luciferase activity, demonstrating the contribution of a translation-independent process. Inhibition of translation suppresses the "fast" component of the EJC decay. From the relative amplitudes of the "slow" translation-independent contributions, one estimates that only 31% of the EJC present in untreated cells at steady-state disassemble following a translation dependent mechanism. Translating ribosomes account for a minor proportion of EJC disassembly after treatment with inhibitors. The translation-dependent EJC disassembly is much faster (17-fold) than the translation-independent one. In untreated growing cells, the EJCs concentration at steady state implies that assembly and disassembly occur at the same rates. Therefore, one deduces that the assembly rate of newly deposited EJC is only 7-fold faster if the complex is prone to be disassembled by the translation-dependent process (Supplementary methods 1 and scheme - Fig. 4d). For that reason, although the majority (69%) of EJCs present in a cell at steady state undergo slow disassembly, most, 85% of newly deposited EJCs are disassembled by the fast, translation-dependent process in agreement with the common thought that EJCs are removed by the pioneer round of translation. To conclude, NanoBiT assays provide a qualitative and quantitative view on the relative contributions of EJC disassembly mechanisms.

The extent of EJC binding is transcript-specific. EJCs have been shown to be deposited −27nt upstream exon junctions on average[30,48]. Furthermore, in Drosophila cells, EJCs appears to be universally deposited on exon junctions[49] but accumulate more on genes containing multiple introns than others[50]. Given the conservation of the splicing machinery and of the EJC core proteins, one might expect the same in mammalian cells. We find that the extent of EJC binding is transcript-specific and poorly related to the amount of transcripts in inputs. Remarkably enough, highly expressed housekeeping gene transcripts such as ribosomal protein coding transcripts, are poorly enriched as previously reported[51]. Yet, eIF4A3 and Y14 have been reported to coordinate ribosome biogenesis, but an indirect effect is considered[52,53]. There remain several possible interpretations for poor enrichment of housekeeping genes: (1) it could be a bias due to their relative abundances and stabilities, newly synthesized transcripts would constitute a small proportion of the total, which would result in elevated unspecific (blank) immunoprecipitation; (2) their relatively low number of introns; (3) while still in the nucleus, some EJCs are disassembled too rapidly to be observable by cross-linking or to be isolated in low salt cytosolic fractions.

The average EJCs persistence with EJC is transcript-dependent. It is over 2 h which is a rather long time. However, we are currently not able to ascribe EJC-transcript persistence at the gene level to translation-dependent or translation-independent pathway. Importantly, persistence of non-coding RNAs is not particularly long. For instance, *SNHG19* shows one of the shortest persistence time (Fig. 6e, Supplementary Data 6). There is a transcript ontology bias in persistence (Supplementary Data 5). Stress granule encoding transcripts are overrepresented among EJC-associated transcripts. More significant, 70 out of 546 most EJC-enriched transcripts and remarkably, 12 among the 30 most persistent EJC-bound transcripts encode proteins linked to centrosomes, microtubules, mitotic spindle, kinetochore, centrioles and ciliogenesis (Supplementary Table 1). The centriole substructure of centrosomes constitutes a basal body for primary cilium formation. Centrosome-nucleated microtubules interact with the kinetochore that assembles on chromosomes to form the mitotic spindle. The connection with EJC is supported by several studies. A highly specific eIF4A3 inhibitor has been shown to interfere with stress granule formation and to arrest the cell cycle at an early stage of mitosis, at the G2/M checkpoint[54]. Deficiencies in *eIF4A3* or *MAGOH* are associated with mitotic delays[55]. EJC protein depletion impairs centrosome organization and ciliogenesis[13]. This work also reported localization of *NIN* mRNA at centrosomes as a unique case of mammalian mRNA reported to requires EJC and translation to localize. *ASPM* (Abnormal Spindle-like Microcephaly-associated) and *PCNT* (pericentrin) coding transcripts included in our EJC-bound transcript list, have both been shown to require an association with polysomes to localize at centrosomes[56,57]. It is tempting to speculate that stable transcript binding to a constrained EJC-ribosome particle might be linked to mRNAs delayed in their initial complete translation, waiting for the appropriate time to be expressed at their destination.

## Methods
A detailed list of reagents, plasmids, antibodies, cells, software and apparatuses is provided in Supplementary Tables 2–7.

### Plasmids and cells
In brief, eIF4A3-SmBiT and MAGOH-LgBiT plasmids are derived from pCDNA3, from pBiT1.1, pBiT2.1 from Promega and from the appropriate human cDNAs. All inserts are checked by sequencing. Human cells are propagated in DMEM supplemented with 10% foetal calf serum. HA-eIF4A3 cells are CRISPR edited HeLa cells in which the HA tag was fused to both eIF4A3 alleles[30]. HEK293 Trex™ (Invitrogen #R) cells are transfected by JET Prime. Stable cell lines are isolated from HEK293 Trex cells after selection with zeocin, puromycin or neomycin.

**Luciferase assays and Western blots.** 100 ng of furimazine, 1 mg/ml stock solution in 50:50 ethanol/propylene glycol) in 30 µl of 0.5× Passive Lysis Buffer (Promega # E1941) per assay are added to 50 µl of lysates, cell suspensions or gradient fractions. The luciferase activity is detected with a Berthold TriStar LB941 luminometer. All cells are propagated in DMEM supplemented with 10% foetal calf serum. For Western blots, primary antibodies were used at 1/1000 dilution and secondary antibodies at 1/5000 dilution. Clean-Blot™ IP Detection Reagent (HRP) (Thermofisher) was used when analyzing immunoprecipitations to avoid background from immunoprecipitating antibodies.

**Kinetic experiments.** For kinetic experiments, OB9 cells (expressing both Flag-eIF4A3-SmBiT and MAGOH-LgBiT) are seeded on polylysine-coated 12-well plates. Prewarmed DMEM medium containing cycloheximide, harringtonine, actinomycin D and/or DRB, is added at pre-determined times prior to lysis. Incubation is arrested on ice while the medium is rapidly sucked off, cells are scraped in 400 µl ice-chilled 0.5× Passive Lysis Buffer per well. Each measure is an average for three

wells (technical replicates) treated independently. Three wells per plate are not exposed to the drugs and their average luciferase activity is used to normalize the data for drug exposed cells on the same plate. Standard deviations are determined from biological replicates obtained on different days.

**Sucrose gradient fractionation.** All buffer solutions are treated with diethyl pyrocarbonate. HEK293 Trex or OB9 cells are seeded on polylysine coated dishes, washed in chilled HKM200 buffer (10 mM Hepes pH7.6, 5 mM $MgCl_2$, 200 mM KCl) and lysed in the same buffer supplemented with 1% NP-40, 1 µl/ml protease inhibitor cocktail (Calbiochem #535140), 40 units/ml rRNAsin (Recombinant RNAsin Ribonuclease Inhibitor, Promega) and 0.5 µM DTT. RNase digestions were performed without RNase inhibitor in the lysis buffer, adding 1 µl/ml RNase A/T1 mix (Thermo Scientific #EN0551) to lysates on ice during at least 30 min. Experiments using 3 times more RNase showed the same results. For ribosome/polysome fractionation, EJC-NanoBiT expressing cells are treated with harringtonine (2 µg/ml) or cycloheximide (100 µg/ml), washed once with ice-cold (10 mM Hepes pH7.6, 5 mM $MgCl_2$, 100 mM KCl) and lysed for 10 min on ice in the same buffer supplemented with NP-40 1%, 1 µl/ml protease inhibitor cocktail (Calbiochem #535140), 40 units/ml rRNAsin and 0.5 µM DTT. Cell lysates clarified by centrifugation 10 min at 4 °C at 16,000 g, are loaded on 11-54% sucrose gradients in 20 mM Hepes pH7.6, 100 mM KCl, 5 mM $MgCl_2$. Ultracentrifugations are performed at 38,000 RPM during 2H00 at 4 °C in a Beckmann SW41 rotor. For low-resolution fractionation, 1 ml fractions are collected manually. For high-resolution fractionation, 48 × 0.25 ml fractions are collected either via an ISCO model UA6 with optical density at 254 nm measures or collected manually and RNA profiles are determined using an INFINITE M NANO* TECAN microplate reader fluorimeter after addition of 50 µl SYBR green 1/10000 solution (BioVision #B1747-5) to 50 µl of each fractions (390 nm excitation, 540 nm emission). Immunoprecipitations in gradient fractions were performed with anti-eIF4A3 antibodies.

**Isolation of EJC-bound RNA.** Biological replicates correspond to experiments performed on different days (pipeline (Fig. 5a)). Hela HA-eIF4A3 cells were used. For each replicate, the same amount of cells split the same day and treated with DRB or not, are used for each time point. 1 ml of cell lysate from two 15 cm dishes is fractionated as 12 × 1 ml fractions on a sucrose gradient, as described above. RNA is isolated from pooled fractions 3 to 12 that contain ribosomes and polysomes. For biological replicates I and II, 12 µ of anti-HA antibody (Sigma #H6908) are added or not to 4 ml of pooled fractions. After 2 h of incubation at 4 °C, 80 µl of protein-A Dynabeads™ (Thermofisher #10008D) are added and the suspension is left rotating overnight at 4 °C. For biological replicates III and IV sets (Supplementary Fig. 1f), lysates from Hela HA-eIF4A3 and their parental HeLa cell are ultracentrifuged. Pooled gradient fractions 3 to 12 were incubated overnight with anti-HA magnetic beads (Life Technologies #88837). For all immunoprecipitations, beads are washed three times in IP350 buffer (20 mM Hepes (pH 7.5), 350 mM KCl, 10 mM $MgCl_2$, 0.5% NP-40). RNAs remaining on beads are phenol-extracted (Trizol Thermofisher Scientific #15596018) and precipitated in ethanol. The dried precipitate is dissolved in 10 µl RNase-free water and sent in dry ice for polyA selection, cDNA library preparation and Illumina sequencing (AZENTA/GENEWIZ Life Sciences). RNA quantities in each sample are evaluated by Q-BiT and/or Agilent Fragment Analyzer.

**RIP-seq data processing—Pseudogenes and mitochondrial transcripts**
For every sample, we sequenced RNAs from pooled gradient fractions (inputs) and from protein A precipitates with (Ab) or without (blk) HA antibody. A total of 20 to 30 million reads were aligned paired-end on Galaxy France server by HISAT2 to a hg38.109 reference human genome where pseudogenes have been masked (nucleotides in their sequences replaced by N)[34]. Resulting bam files are analysed on Galaxy France server by featureCounts v2.3 with the paired end counted as a single fragment. A count matrix for all samples was set up. To take into account the sequencing depth, we determined reads per million (RPM) for each genes dividing the corresponding read counts by the sum of counts of all aligned reads. For replicates I and II, a matrix is set up with RPM determined from inputs backgrounds and immunoprecipitates from untreated or DRB-treated cell lysates of the two biological replicates (Supplementary Data 1). For each gene (g), we determine RPM for anti-HA immunoprecipitated RNA ($Ab(g)$) and for blank RNA ($blk(g)$) precipitated by protein A in the absence of antibodies. For reliability, we only keep those genes that have at least 50 RPM reads on average of both replicates HA-immunoprecipitated RNAs from untreated cells ($AbO(g)$>50), 4326 genes satisfy this requirement. A gene ontology term enrichment analysis was performed on the genes enriched more than 1-fold, $AbO(g)/blkO(g)$>1 (3110 genes), or more than 4-fold, $AbO(g)/blkO(g)$>4 (546 genes) using AmiGO 2 [https://amigo.geneontology.org/amigo] (Supplementary Data 5).

**Use of mitochondrial encoded transcripts for normalization of count numbers**
We first tested the reproducibility of normalized counts (RPM). For each treatment conditions, there are good correlations between blank precipitated counts $blk(g)$ from replicate (I) with replicate (II). The frequency distribution of $\log_2(blk(g)_I / blk(g)_{II})$ follows a normal (Gaussian) distribution as expected for unspecific random binding to protein A beads. In contrast, the frequency distribution of $\log_2(Ab(g)_{II}/blk(g)_{II})$ as well as $\log_2(Ab(g)_I/ blk(g)_I)$ ratios markedly deviate from a normal distribution and are more dispersed between replicates than $(blk(g)_I/ blk(g)_{II})$ (Supplementary Fig. 1c). In total 3110 out of 4326, about 3/4 of the genes, show $Ab(g)/blk(g)$ > 1 on average for both replicates. Thus, a very high proportion of transcripts are specifically immunoprecipitated with EJC.

The "true" number of EJC-bound transcripts ($Ejc(g)$) must take into account a background contribution due to unspecific RNA precipitation. Since a major proportion of transcripts are immunoprecipitated, a normalization procedure based on definition of invariant genes (g') is required[35]. The blank contribution $blk(g)$ should be weighted by a factor, α. $Ejc(g) = Ab(g) - α\ blk(g)$ and for invariant genes, $Ejc(g°) = 0 = Ab(g°) - α\ blk(g°)$.

Mitochondrial-encoded transcripts are attractive candidates to serve as invariants. They constitute 5–10% of mapped reads both in inputs and immunoprecipitates. They are not spliced, therefore highly unlikely to associated with an EJC. $Ab(mt-g)$ are plotted versus $blk(mt-g)$ and the slopes of the linear correlations provide an estimate of α for every replicate (Supplementary Data 3). This estimate is used next to evaluate the "true" number of EJC-bound transcripts ($Ejc(g)$) (Supplementary Data 4).

Persistence indexes, $Ejc120(g)/EjcO(g)$ and $Ejc60(g)/EjcO(g)$ ratios, are calculated independently for each replicate (Supplementary Data 6). They are next normalized by the median of ratios normalization method and averaged. To limit the impact of blank contributions, we focussed on the 546 genes where $Ab(g)/blk(g)$ > 4 on average.

## Data availability
The high throughput RNA sequencing data are available in the Sequence Read Archive (SRA) database: PRJNA1002469. Excel files and Western blots used for figures are available as Source Data file. Supplementary Data files as well as Source Data files are available at Figshare [https://doi.org/10.6084/m9.figshare.25511059]. Integrative Genome Viewer images will be provided on request. The PDB data 2J0S was used to generate Fig. 1a. Source data for figures and Supplementary figures are provided with this paper. Source data are provided with this paper.

## Material availability

Correspondence and requests should be addressed to Hervé Le Hir (lehir@bio.ens.psl.eu) or Olivier Bensaude (bensaude@bio.ens.psl.eu).

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

## Acknowledgements

This work was initiated while O.B. was a Weston visiting professor at the Weizmann Institute of Science. We thank all HLH lab members for fruitful discussions, Cindy Burel, Pr. Alice Lebreton, Pr. John T. Lis and Elias Marzoner for critical reading of the manuscript, Alexandra Louis for masking pseudogenes on a human reference genome, Nikita Menezes Bragança for making the figures. This work was supported by the Agence Nationale de la Recherche (ANR-17-CE12-0021 and ANR-21-CE12-0041) (H.L.H.), Fondation pour la Recherche Médicale (FRM-EQU202003010226) (H.L.H.), programme Investissements d'Avenir [ANR-10-LABX-54 MEMOLIFE and ANR-10-IDEX-0001-02 PSL* Research University to L.M. and H.L.H.] (H.L.H.) and by continuous financial support from the Centre National de Recherche Scientifique, the Ecole Normale Supérieure and the Institut National de la Santé et de la Recherche Médicale, (H.L.H.) as well as by Weizmann Institute internal grants from the Estate of Manfred and Margaret Tannen and the Joel and Mady Dukler Fund for Cancer Research (R.D.).

## Author contributions

O.B. and I.B. performed experiments, O.B. wrote the paper, L.M. performed bioinformatics analysis, R.D. supported starting the project, H.L.H. discussed the results, managed the team and secured funding.

## Competing interests

The authors declare no competing interests.
