## [Peer Review File · Nature Communications]

REVIEWER COMMENTS

Reviewer #1 (Remarks to the Author):

In this interesting manuscript, the Le Hir lab carries investigates the kinetics of Exon Junction complex (EJC) assembly and disassembly.

The experimental approach involves the use of human cells overexpressing EJC core subunits tagged with complementary nanoluciferase fragments (NanoBiT). This protein complementation assay was used to test the interactions of MAGOH and Y14 and of eIF4A3 with MAGOH. They also used the NanoBiT system to test mutations that have been shown to impair EJC assembly. From these studies the authors concluded that the interaction of MAGOH-C-LgBiT and eIF4A3-C-SmBiT can be used as diagnostic of EJC assembly.

Next, Bensaude and colleagues used the NanoBiT system to investigate the dependency on mRNA translation for EJC assembly and disassembly. They fractionated EJC complexes on sucrose gradients and observed that EJC form large mRNPs particles broadly distributed over 100S, which include stalled ribosomes. Interestingly, the authors found that most of the EJCs undergo a slow translation-independent disassembly mechanism, as compared to 30% of EJCs that rely on a fast translation-dependent mechanism. This is a very important finding.

Finally, the authors use RIP-seq to investigate the transcript dependence of EJC assembly and disassembly. They could not find any defining features related to transcript binding to EJC (e.g., transcript length, RNA stability, etc.). However, they observed that the lifetime of EJC particles is clearly transcript-specific. They noticed that EJC-associated transcripts are enriched in housekeeping genes, in particular mRNA encoding proteins linked to centrosomes, microtubules and/or ciliogenesis.

In summary, this work is of general interest since it sheds light on the kinetics of EJC assembly and disassembly. The experiments are well designed, and the conclusions are supported by the data.

Specific comments

- Almost exclusively all conclusions on Figures 1-4 are based on the split luciferase assay. Having said that, all the experiments seem robust, and the proper controls have been included. Nonetheless, it

would add a new dimension, if some of the results could be validated using an additional approach that does not involve protein complementation assay.

- The transcript-specificity of EJC is an important finding. However, the authors acknowledge on page 13 “Whether EJCs are deposited on all exon junctions during splicing remains an open question”. Thus, the authors should tone down the emphasis on a link between EJC lifetime and a link to microtubules and centrosomes.

Minor

- The cartoon displayed in Fig 1b should show the N and C-term, to make it clear that the fusion is between the N-term of MAGOH with the C-term of Y14, as explained in the text.

- On Fig 1f and 1g, the authors should explain the effect of the MAGOH and eiF4E3 mutations in EJC assembly, as they do for the T163A/D mutations, and not only refer to a previous publication

Reviewer #2 (Remarks to the Author):

In this manuscript the authors characterize the assembly and disassembly kinetics of the exon junction complex. They use the NanoBiT system to quantitate the EJC but importantly confirm key results by detecting endogenous EJCs by co-IP. They also study where in the transcriptome EJCs are that are relatively stable by RIPseq and show that some EJC/mRNA complexes are stable even 2 hours after transcription is inhibited whereas others disappear rapidly after transcription is inhibited. Overall the experiments are interesting and well controlled. The authors make some very important observations, but I think the manuscript could be improved with some additional analysis of the data.

Major points:

1. The importance of fast, translation-dependent EJC disassembly may be severely underestimated by the authors. If the slowly disassembled EJCs are 17 times as stable but only twice as abundant at steady state as the unstable EJCs, the unstable EJCs must be made/deposited at a much higher rate. My back of

the envelope calculations suggest that about 90% of all newly deposited EJCs must be of the rapidly disassembled type to account for their 1/3 abundance at steady state. Can the authors comment?

2. The RIP-seq approach focuses on annotated genes, which are enriched for protein coding transcripts. Many ncRNAs are also spliced and presumably get EJCs deposited. This includes the snoRD86 transcript that is highlighted in some of the RIPseq figures. Have authors considered analyzing EJC by transcript (instead of by gene)? This may allow analyzing EJCs in coding transcripts separately from those in noncoding transcripts. Alternatively, can the authors determine whether ncRNAs are more abundant in the EJC fraction 2 hours after transcription inhibition?

Reviewer #3 (Remarks to the Author):

Summary:

In this manuscript, Bensaude and colleagues investigated the kinetics of EJC disassembly from mRNAs using a split luciferase complementation assay (NanoBiT) and RIP-seq. Their results, overall, indicate that only a minority of EJCs undergo fast, translation-dependent disassembly, whereas the majority undergoes slow, translation-independent disassembly. This is a highly interesting and important finding, which challenges the currently prevailing assumption that the first round of translation strips of all EJCs located in an mRNA's coding sequence.

The authors considered the proximity between the eIF4E3 C-term and the MAGOH C-term to be a signal for an assembled EJC and fused SmBiT and LgBiT to the C-termini of these two proteins and confirmed with MAGOH and eIF4A3 mutants known to impair EJC formation that they lead to no or greatly reduced NanoBiT signal (Fig. 1). They then fractionated cell lysates on sucrose gradients and found that the NanoBiT signal peaked in fractions that correspond to densities higher than monosomes. eIF4A3-RIP of these heavy fractions revealed the presence of rRNA, indicating that a large portion of the EJC-associated mRNPs contain one or more ribosomes, which would explain their presence in the higher density fractions (Fig. 2). The authors then treat their lysates with RNases and/or 600 nM NaCl before fractionating them on sucrose gradient. While high ionic strength alone does not alter much the distribution of the NanoBiT signal along the gradient (Fig. 3b), and RNase treatment alone does not release a major fraction of the EJCs from high-density fractions (Fig. 3d), the combination of both treatments causes a major redistribution of the NanoBiT signal (fig. 3a) and co-IP of eIF4A3 with Y14 and MAGOH from the heavy to the light fractions (Fig. 3d), indicating separation of EJCs from ribosomes.

Moreover, the combination of RNase and high salt causes an almost 10-fold increase in the NanoBiT signal, suggesting that at low ionic strength, luciferase activity is somehow inhibited (Fig. 3c).

The authors then attempted to estimate the kinetics of EJC assembly and disassembly kinetics by applying a simple kinetic model that relies on several not tested assumptions, for example that transcription or splicing (i.e. the EJC assembly rates) are not affected by translation inhibition for 2.5 hours. Under these assumptions, they postulate that only 30% of the EJC disassembly depends on translation (Fig. 4a). They bolster this conclusion by measuring EJC disassembly in time course experiments under conditions where transcription was inhibited and show that about 30% of the EJCs disassemble fast and in a translation-dependent manner (sensitive to CHX), while the rest disassembles 15 times slower and independent of ongoing translation (Fig. 4b-d). On the other hand, the large difference in EJC assembly rates after transcription re-start, dependent on whether translation is on or off, seems to indicate that translation contributes to a major fraction of EJC disassembly.

Sequencing of poly(A)+ RNA co-immunoprecipitated with HA-eIF4A3 (RIP-seq) revealed 546 genes whose RNA was > 4-fold enriched in the IP compared to the input. From my understanding of this elaborate RIP-seq approach (incl. time course and transcription arrest), the take-home message is that there is no correlation between RNA half-life and EJC persistency on RNAs. Both are highly transcript-specific and no determinants/rules for EJC persistency could be found.

Overall, I find the development of the NanoBiT assay for assessing the dynamics of EJC complexes highly original, elegant, and useful. Moreover, the presented results of this assay are unexpected, against the common assumption that most RNA-bound EJCs are displaced from CDS by elongating ribosomes, and therefore interesting for the research community. However, the manuscript falls short to provide an explanation for this unexpected finding, and the RIP-seq data presented in Figs. 5 and 6 does also not provide any further insight or an explanation.

Points to address:

Fig. 1d-g: No error bars are shown and no information regarding replicates is given in the figure legend. Does this data represent measurements from one single experiment? It would be prudent to show an average \pm SD of 3 replicate experiments here.

Comparison of Figs. 2d and e: The distribution of the NanoBiT signal is similar in the gradients from CHX- and harringtonine-treated cells, which should be discussed given that harringtonine enriches for mRNPs with one ribosome loaded (at the start codon), whereas CHX enriches for mono and polysomes alike.

Wouldn't one expect that in harringtonine-treated cells, the signal would drop sharply after the 80S peak, assuming that ribosomes rather than EJCs or other mRNP factors determine where an mRNP migrates in these sucrose gradients?

The data presented in Fig. 3 is difficult to interpret, as it is not entirely clear how increased ionic strength and RNase treatment exactly affect mRNPs. Ionic strength for example may alter migration behavior in sucrose gradients through changes in mRNP composition that are independent of EJCs or ribosomes. Moreover, RNase treatment alone appears to only inefficiently separate EJCs from ribosomes, which might suggest an overall compact mRNP structure in which most of the RNA is not accessible for RNases. The authors should explain better what they mean by "these particles appear to be constrained" (that applies also to the respective statement in the abstract).

Fig. 4: The conclusion that only 30% of the EJCs undergo translation-dependent disassembly in Fig. 4a is based on the untested assumptions that translation inhibition for up to 2.5 hours would not affect EJC assembly rates. However, if the EJC assembly rate was moderately reduced by translation inhibition – which I find an equally plausible assumption –, this would drastically increase the estimated proportion of translation-dependent EJC disassembly. The author's conclusion lacks therefore compelling experimental support. While the data from DRB or ActD treated cells (Figs. 4 b-d) supports the conclusion that about 30% of the EJCs undergo rapid translation-dependent disassembly (this data I find compelling), the transcription block/release experiments (Fig. 4e) show a large increase in EJC assembly rates when translation (and therewith fast EJC disassembly) is inhibited, which would argue that a major fraction of EJCs disassembles in a translation-dependent manner. Unfortunately, the authors do not provide mathematical modeling of this data, arguing that poorly defined delays in transcription recovery do prevent such an analysis. Nevertheless, the authors should discuss the seemingly contradictory data and also attempt the modeling of the data shown in Fig. 4e.

In the description of the RIP-seq data, the authors mention that no correlation between eIF4A3 co-precipitation and gene length, transcript length, or transcript stability was found. How about a correlation between exon number per transcript/gene and enrichment? Another interesting aspect would be to analyze whether lncRNAs are on average more associated with EJCs (and for a longer time) than mRNAs and whether translation inhibition slows down the EJC disassembly rate of mRNAs but not of lncRNAs.

P. 10: The experiment description leading to the data shown in Figure 6 (1st paragraph) is very technical and hard to follow and understand. A more reader-friendly explanation of what was exactly done and why, would be highly appreciated. Furthermore, the description of the selected examples (Fig. 6c-g) lacks any information regarding the rationale for showcasing these specific genes and what we can learn from these examples. I may have not completely grasped what this data overall shows, but to me it seems that the only information it provides is that both, transcript stability and EJC persistence on an RNA, are highly transcript-specific (with no general principles that could be deciphered) and that there is no

correlation between the two. If this is correct, then the entire RIP-seq experiment was disappointingly non-informative. If there is more than that to be learned, the authors should discuss it.

It is also possible that the data has not yet been analyzed thoroughly enough to reveal some general principles. For example, have the authors intersected their enrichment values e.g. with translation efficiencies of the corresponding genes (e.g. ribosome profiling data)?

P. 11: Confusing: You first state that harringtonine should lead to mRNPs with one stalled ribosome located at the initiation codon and then say: "An increase in EJC particle size well above 80S indicates that more than one ribosome remains stalled on the transcripts". So how would you then interpret your data with harringtonine treated cells? You would expect only EJC-associated mRNPs that migrate into the gradient around the 80S peak but not further down.

P. 13: Wouldn't a 4th and probably most obvious alternative explanation for the poor EJC enrichment on housekeeping genes be that they are undergoing very efficient translation, as soon as they enter the cytoplasm, which together with the notion that they tend to have a long half-life, would result in a low proportion of EJC-bound transcripts at any given time?

In the last sentence of the Discussion, the authors finally speculate that the transcripts exhibiting persistent EJC enrichment might be those in which translation is delayed and dependent on a regulatory signal and/or their transport to the right locale in the cell. I find this an obvious hypothesis that could and should be tested by comparing the RIP-seq data with ribosome profiling data (see comment above).

Minor points:

Legend to Fig. 2: Please improve clarity. Dotted and solid lines can hardly be distinguished (panels a, d, and e) and the sentence "RNA in 2A is detected by SYBR green fluorescence, bit in 2a, 2d and 2e ..." does not make sense. Please correct it.

P. 6: A brief description of the assay would facilitate the reader to understand that the indicated times refer to the times during which the cells were incubated with the drugs before they were lysed and NanoBit activity was measured.

Fig. 4c: Translate "ActD ou DRB" into English.

RIP-seq: Please indicate in Fig. 5a which fractions from the sucrose gradient were pooled for the RIP-seq.

P. 9: Please correct: “We could not find detect any straightforward ...”

Figs. 6a and 6b should be swapped since panel b is referred to in the text earlier than panel a.

P. 10: Please clarify the following sentence: “Overall, genes that are most affected by transcriptional arrest of 2 hours are the most affected by 1 hour.” Affected by 1 hour of what?

P. 10: “neighboring genes genes immunoprecipitated ...”, delete one of the “genes”.

P. 11, 2nd paragraph: “Therefore” instead of “Therefre”

We thank the reviewers for their useful comments and have modified the manuscript accordingly.

RESPONSES to REVIEWER COMMENTS

Reviewer #1 (Remarks to the Author):

In this interesting manuscript, the Le Hir lab carries investigates the kinetics of Exon Junction complex (EJC) assembly and disassembly. The experimental approach involves the use of human cells overexpressing EJC core subunits tagged with complementary nanoluciferase fragments (NanoBiT). This protein complementation assay was used to test the interactions of MAGOH and Y14 and of eIF4A3 with MAGOH. They also used the NanoBiT system to test mutations that have been shown to impair EJC assembly. From these studies the authors concluded that the interaction of MAGOH-C-LgBiT and eIF4A3-C-SmBiT can be used as diagnostic of EJC assembly.

Next, Bensaude and colleagues used the NanoBiT system to investigate the dependency on mRNA translation for EJC assembly and disassembly. They fractionated EJC complexes on sucrose gradients and observed that EJC form large mRNPs particles broadly distributed over 100S, which include stalled ribosomes. Interestingly, the authors found that most of the EJCs undergo a slow translation-independent disassembly mechanism, as compared to 30% of EJCs that rely on a fast translation-dependent mechanism. This is a very important finding.

Finally, the authors use RIP-seq to investigate the transcript dependence of EJC assembly and disassembly. They could not find any defining features related to transcript binding to EJC (e.g., transcript length, RNA stability, etc.). However, they observed that the lifetime of EJC particles is clearly transcript-specific. They noticed that EJC-associated transcripts are enriched in housekeeping genes, in particular mRNA encoding proteins linked to centrosomes, microtubules and/or ciliogenesis.

In summary, this work is of general interest since it sheds light on the kinetics of EJC assembly and disassembly. The experiments are well designed, and the conclusions are supported by the data.

Specific comments

- Almost exclusively all conclusions on Figures 1-4 are based on the split luciferase assay. Having said that, all the experiments seem robust, and the proper controls have been included. Nonetheless, it would add a new dimension, if some of the results could be validated using an additional approach that does not involve protein complementation assay.

Gradient profiles obtained with split-luciferase (Figure 2a) are consistent with profiles obtained by immunoprecipitation of wild-type endogenous EJC with an eIF4A3 antibody (Figure 2b). Data obtained from RIP-sequencing of EJC bound transcripts after transcriptional arrest are consistent with the conclusion of split-luciferase assays that

the average persistence of transcripts with EJC is rather long (several hours) (Figure 6 and Supplemental data file 6).

- The transcript-specificity of EJC is an important finding. However, the authors acknowledge on page 13 “Whether EJCs are deposited on all exon junctions during splicing remains an open question”.

We now quote on page 13, a recent paper from our lab (new reference 17) that shows that in *Drosophila* cells, EJC are deposited universally on exon-exon junctions. This is likely also the case for human cells given the conservation of the splicing machinery and of the EJC core proteins.

Thus, the authors should tone down the emphasis on a link between EJC lifetime and a link to microtubules and centrosomes.

We agree that the link between EJC lifetime and microtubules or centrosomes is weak. Yet even short lifetimes are relatively long. We now write (in the abstract) that “RIP sequencing showed a bias of EJC bound transcripts towards microtubule and centrosome coding ones and demonstrated that EJC particle lifetimes are transcript-specific”

Minor

- The cartoon displayed in Fig 1b should show the N and C-term, to make it clear that the fusion is between the N-term of MAGOH with the C-term of Y14, as explained in the text. Figure 1b has been modified accordingly.

- On Fig 1f and 1g, the authors should explain the effect of the MAGOH and eIF4E3 mutations in EJC assembly, as they do for the T163A/D mutations, and not only refer to a previous publication

We now write “MAGOH residues K41 to D43 form contact with two domains of eIF4A3 (ref 20).... The eIF4A3 DE401/402KR mutation abolishes a salt bridge between eIF4A3 and Y14 (ref 19, 20)”.

Reviewer #2 (Remarks to the Author):

In this manuscript the authors characterize the assembly and disassembly kinetics of the exon junction complex. They use the NanoBiT system to quantitate the EJC but importantly confirm key results by detecting endogenous EJCs by co-IP. They also study where in the transcriptome EJCs are that are relatively stable by RIPseq and show that some EJC/mRNA complexes are stable even 2 hours after transcription is inhibited whereas others disappear rapidly after transcription is inhibited. Overall the experiments are interesting and well controlled. The authors make some very important observations, but I think the manuscript could be improved with some additional analysis of the data.

Major points:

1. The importance of fast, translation-dependent EJC disassembly may be severely

underestimated by the authors. If the slowly disassembled EJCs are 17 times as stable but only twice as abundant at steady state as the unstable EJCs, the unstable EJCs must be made/deposited at a much higher rate. My back of the envelope calculations suggest that about 90% of all newly deposited EJCs must be of the rapidly disassembled type to account for their 1/3 abundance at steady state. Can the authors comment?

After one hour of transcription (hence splicing) arrest, 50% of EJCs remain (Figure 4b). This demonstrates that less than 50 % are rapidly disassembled. When both translation and transcription are arrested, 70% of EJCs remain after one hour (Figure 4c). This is in agreement with 30% of EJC rapidly (translation-dependent) disassembled. Furthermore, rate constants, k_{off} estimated from transcription inhibition (Figures 4b and 4c) strikingly account quantitatively for the translation inhibition effect (Figure 4a).

2. The RIP-seq approach focuses on annotated genes, which are enriched for protein coding transcripts. Many ncRNAs are also spliced and presumably get EJCs deposited. This includes the snoRD86 transcript that is highlighted in some of the RIPseq figures. Have authors considered analyzing EJC by transcript (instead of by gene)? This may allow analyzing EJCs in coding transcripts separately from those in noncoding transcripts.

Indeed, an analysis per transcript rather than per gene would provide additional clues. Unfortunately, such analysis is not possible here since Illumina sequencing only provides short reads (150-200 nt on average).

Alternatively, can the authors determine whether ncRNAs are more abundant in the EJC fraction 2 hours after transcription inhibition?

Several ncRNA are found associated with EJC. SNHG (1, 3, 4, 12, 19) and LINC00173 and enriched more than 4-fold (see Supplemental data file 4). SNHG.EJC tend to be less persistent on average than other transcripts. SNHG19 shown in Figure 6a is typical. The case of snoRD86 is uniquely relevant for one of the NOP56 alternative transcripts which contains it and has been extensively investigated in reference (37). We have looked at other sno-containing transcripts, they do not show any specific behavior.

Reviewer #3 (Remarks to the Author):

Summary:

In this manuscript, Bensaude and colleagues investigated the kinetics of EJC disassembly from mRNAs using a split luciferase complementation assay (NanoBiT) and RIP-seq. Their results, overall, indicate that only a minority of EJCs undergo fast, translation-dependent disassembly, whereas the majority undergoes slow, translation-independent disassembly. This is a highly interesting and important finding, which challenges the currently prevailing assumption that the first round of translation strips of all EJCs located in an mRNA's coding sequence.

The authors considered the proximity between the eIF4E3 C-term and the MAGOH C-term to be a signal for an assembled EJC and fused SmBiT and LgBiT to the C-termini of these two proteins and confirmed with MAGOH and eIF4A3 mutants known to impair

EJC formation that they lead to no or greatly reduced NanoBiT signal (Fig. 1). They then fractionated cell lysates on sucrose gradients and found that the NanoBiT signal peaked in fractions that correspond to densities higher than monosomes. eIF4A3-RIP of these heavy fractions revealed the presence of rRNA, indicating that a large portion of the EJC-associated mRNPs contain one or more ribosomes, which would explain their presence in the higher density fractions (Fig. 2). The authors then treat their lysates with RNases and/or 600 nM NaCl before fractionating them on sucrose gradient. While high ionic strength alone does not alter much the distribution of the NanoBiT signal along the gradient (Fig. 3b), and RNase treatment alone does not release a major fraction of the EJCs from high-density fractions (Fig. 3d), the combination of both treatments causes a major redistribution of the NanoBiT signal (fig. 3a) and co-IP of eIF4A3 with Y14 and MAGOH from the heavy to the light fractions (Fig. 3d), indicating separation of EJCs from ribosomes. Moreover, the combination of RNase and high salt causes an almost 10-fold increase in the NanoBiT signal, suggesting that at low ionic strength, luciferase activity is somehow inhibited (Fig. 3c).

The authors then attempted to estimate the kinetics of EJC assembly and disassembly kinetics by applying a simple kinetic model that relies on several not tested assumptions, for example that transcription or splicing (i.e. the EJC assembly rates) are not affected by translation inhibition for 2.5 hours. Under these assumptions, they postulate that only 30% of the EJC disassembly depends on translation (Fig. 4a). They bolster this conclusion by measuring EJC disassembly in time course experiments under conditions where transcription was inhibited and show that about 30% of the EJCs disassemble fast and in a translation-dependent manner (sensitive to CHX), while the rest disassembles 15 times slower and independent of ongoing translation (Fig. 4b-d). On the other hand, the large difference in EJC assembly rates after transcription re-start, dependent on whether translation is on or off, seems to indicate that translation contributes to a major fraction of EJC disassembly.

Sequencing of poly(A)⁺ RNA co-immunoprecipitated with HA-eIF4A3 (RIP-seq) revealed 546 genes whose RNA was > 4-fold enriched in the IP compared to the input. From my understanding of this elaborate RIP-seq approach (incl. time course and transcription arrest), the take-home message is that there is no correlation between RNA half-life and EJC persistency on RNAs. Both are highly transcript-specific and no determinants/rules for EJC persistency could be found.

Overall, I find the development of the NanoBit assay for assessing the dynamics of EJC complexes highly original, elegant, and useful. Moreover, the presented results of this assay are unexpected, against the common assumption that most RNA-bound EJCs are displaced from CDS by elongating ribosomes, and therefore interesting for the research community. However, the manuscript falls short to provide an explanation for this unexpected finding, and the RIP-seq data presented in Figs. 5 and 6 does also not provide any further insight or an explanation.

Points to address:

Fig. 1d-g: No error bars are shown and no information regarding replicates is given in

the figure legend. Does this data represent measurements from one single experiment? It would be prudent to show an average \pm SD of 3 replicate experiments here.

We now write in the figure legend that biological triplicates were performed. Error bars are now shown.

Comparison of Figs. 2d and e: The distribution of the NanoBiT signal is similar in the gradients from CHX- and harringtonine-treated cells, which should be discussed given that harringtonine enriches for mRNPs with one ribosome loaded (at the start codon), whereas CHX enriches for mono and polysomes alike.

Wouldn't one expect that in harringtonine-treated cells, the signal would drop sharply after the 80S peak, assuming that ribosomes rather than EJCs or other mRNP factors determine where an mRNP migrates in these sucrose gradients?

We now write on page 5 :” Harringtonine does not inhibit translation elongation and termination but stalls ribosomes at initiation codons resulting in 80S mRNP accumulation on translating mRNAs. Therefore, a EJC ribosome particle size remaining higher, indicates that its ribosomes are stalled. As both cycloheximide and harringtonine can favour reinitiation adding an extra ribosome to EJC particles ³¹, the observed increased EJC particle size might be attributed to additional loading of a ribosome” and page 12 that : “In the presence of harringtonine, elongating ribosomes are expected to terminate normally and drop off. Reinitiating ribosomes arrest as 80S particles. Therefore an EJC particle size remaining even increasing well above 80S, demonstrates that its ribosomes are not elongating translation, they are stalled. Both cycloheximide and harringtonine may favour reinitiation adding an extra ribosome to EJC particles ³¹. The increased in EJC particle size would be due to additional loading of ribosomes stalling at the initiation codon (harringtonine) or are frozen after initiation (cycloheximide).”

The data presented in Fig. 3 is difficult to interpret, as it is not entirely clear how increased ionic strength and RNase treatment exactly affect mRNPs. Ionic strength for example may alter migration behavior in sucrose gradients through changes in mRNP composition that are independent of EJCs or ribosomes.

We agree that RBPs other than ribosomes are present on mRNAs with EJCs may be affected by ionic strength and RNase. However, as discussed on page 12, ribosomes (around 4,000 kDa) are likely to be major contributors to EJC (350 kDa) particle sizes from 80S and over (page 12).

Moreover, RNase treatment alone appears to only inefficiently separate EJCs from ribosomes, which might suggest an overall compact mRNP structure in which most of the RNA is not accessible for RNases.

We fully agree with this statement. We have reworded the first paragraph in the “Increasing ionic strength relieves constraints in EJC-ribosome particles” chapter, page 5:” Increasing ionic strength weakens intermolecular ionic interactions. Addition of 600 mM NaCl to the lysates slightly shifts the luciferase peak towards smaller particles at the top of gradients (Figure 3a). Large EJC particles resist high ionic strength. Are EJC and ribosomes linked by an RNA molecule? When NanoBiT EJC lysates are digested with a mixture of RNases A and T1, a luciferase peak appears towards the top of the gradient

(Figure 3b). But a large amount of NanoBiT EJC particles remains larger than 80S. This profile remains identical when using three times higher concentrations of RNases eliminating a kinetic effect to account for the incomplete transformation of the large complexes into small ones (data not shown). But when lysates are supplemented with both 600 mM NaCl and RNase, the large NanoBiT EJC complexes disappear completely and are replaced by a peak at the top of the gradients (Figure 3b). When ionic strength is increased, RNA becomes accessible to RNases. Note that 80S ribosomes resist RNase treatments at high salt. The RNase effect strongly suggests that transcripts link EJC to ribosomes.”

The authors should explain better what they mean by “these particles appear to be constrained” (that applies also to the respective statement in the abstract).

The idea of “constraint” is suggested by ionic strength and RNase effects on split-luciferase activity. We reworded the abstract :” Most of these particles contain stalled ribosomes. They are constrained as increasing ionic strength results in enhanced susceptibilities to RNase and split-luciferase activities.” In discussion page 12 we write: “To match properly, movements of the split luciferase fragments should not be sterically impaired by neighbor macromolecules. Ionic strength and RNase digestion weaken protein interactions therefore alleviate constraints preventing efficient split-luciferase matching..”

Fig. 4: The conclusion that only 30% of the EJCs undergo translation-dependent disassembly in Fig. 4a is based on the untested assumptions that translation inhibition for up to 2.5 hours would not affect EJC assembly rates. However, if the EJC assembly rate was moderately reduced by translation inhibition – which I find an equally plausible assumption –, this would drastically increase the estimated proportion of translation-dependent EJC disassembly. The author’s conclusion lacks therefore compelling experimental support.

We now write on page 6 “Transcription and splicing rates are assumed not be significantly affected at least during a few hours of treatment, EJC assembly rates are expected to remain constant. To maximize the validity of this assumption, we limited our experiments to 150 minutes of treatment.”

The striking consistency between time constants obtained independently from translation inhibition effects and from transcription inhibition (Figure 4a, b, c, d) strongly supports (but does not prove) the assumption that translation inhibition does not affect EJC assembly rates. Actually, long translation inhibition times (over several hours) may lead to slightly decreased assembly rates (not shown).

While the data from DRB or ActD treated cells (Figs. 4 b-d) supports the conclusion that about 30% of the EJCs undergo rapid translation-dependent disassembly (this data I find compelling), the transcription block/release experiments (Fig. 4e) show a large increase in EJC assembly rates when translation (and therewith fast EJC disassembly) is inhibited, which would argue that a major fraction of EJCs disassembles in a translation-dependent manner. Unfortunately, the authors do not provide mathematical modeling of this data, arguing that poorly defined delays in transcription recovery do prevent such

an analysis. Nevertheless, the authors should discuss the seemingly contradictory data and also attempt the modeling of the data shown in Fig. 4e.

We now write on page 8 that “If DRB-containing medium is replaced by cycloheximide-containing medium, the recovery is 2-fold faster. We expected 1.4 fold according to Figures 4a-d. When transcriptional stress is released, specific transcripts might be transcribed and these transcripts might be more susceptible than steady-state ones to translation-dependent EJC removal.”

In the description of the RIP-seq data, the authors mention that no correlation between eIF4A3 co-precipitation and gene length, transcript length, or transcript stability was found. How about a correlation between exon number per transcript/gene and enrichment? Another interesting aspect would be to analyze whether lncRNAs are on average more associated with EJCs (and for a longer time) than mRNAs and whether translation inhibition slows down the EJC disassembly rate of mRNAs but not of lncRNAs.

In page 10, we now write :” We could not detect any straightforward features related to transcript binding to the EJC : No significant links to exon number, gene length, transcript length, or stability. Several non-coding RNAs such as SNHG (1, 3, 4, 12, 19) and LINC00173 are enriched more than 4-fold (see Supplemental data file 4).“ In page 13 : “Importantly, the persistence of non coding RNAs is not particularly longer. For instance, SNHG19 shows one of the shorter persistence time (Figure 6e). “

P. 10: The experiment description leading to the data shown in Figure 6 (1st paragraph) is very technical and hard to follow and understand. A more reader-friendly explanation of what was exactly done and why, would be highly appreciated.

We now write : “The decrease in Ejc(g) following an arrest in transcription hence EJC assembly, reflects the persistence of transcript (g) with the EJC.” “On average, Ejc⁰(g) is larger than Ejc⁶⁰(g) that is in itself larger than Ejc¹²⁰(g) which is to be expected since no new EJCs assemble (Figure 6a).”

Furthermore, the description of the selected examples (Fig. 6c-g) lacks any information regarding the rationale for showcasing these specific genes and what we can learn from these examples.

“In Figure 6c-h, we show data for 6 individual transcripts illustrating the various behaviours observed. As written in pages 10-11, “two extreme classes of EJC-bound transcripts are distinguishable: short-lived ones such as *PNRC2* that strongly decrease in immunoprecipitates and inputs upon DRB treatment (Figure 6c) and persistent ones such as *MIIP* that do not decrease in either inputs or immunoprecipitates following an arrest in transcription (Figure 6d). Some transcripts such as *SNHG19* are non coding, strongly decrease in immunoprecipitates but remain stable in inputs (Figure 6e). EJC enrichment in alternative *CIRBP* transcripts (red stars) decreases faster after DRB than that of other transcripts thereby illustrating a differential persistence time within transcripts coded by the same gene (Figure 6f). *SRSF7* provides an example of alternative transcript enriched with EJC but all transcripts decrease at a similar rate (Figure 6g). Furthermore, *NOP56* is an extreme case where alternative transcript/EJC association is very stable (Figure 6h). “

I may have not completely grasped what this data overall shows, but to me it seems that the only information it provides is that both, transcript stability and EJC persistence on an RNA, are highly transcript-specific (with no general principles that could be deciphered) and that there is no correlation between the two. If this is correct, then the entire RIP-seq experiment was disappointingly non-informative. If there is more than that to be learned, the authors should discuss it.

Indeed, it is unfortunate, that the current data essentially demonstrates that EJC persistence on an RNA, is highly transcript-specific. Further studies are required and planned to establish rules for disassembly rates which we find an extremely interesting question.

It is also possible that the data has not yet been analyzed thoroughly enough to reveal some general principles. For example, have the authors intersected their enrichment values e.g. with translation efficiencies of the corresponding genes (e.g. ribosome profiling data)?

“Inputs” from which EJCs are immunoprecipitated are pooled 80S and polysome fractions. Importantly, several non coding RNAs are strongly enriched in these fractions. Yet they are unlikely to be translated. This been said, ribosome footprinting should and will be performed in future studies to investigate the positioning of ribosomes on EJC-bound transcripts. Are they stalled at random or at specific positions?

P. 11: Confusing: You first state that harringtonine should lead to mRNPs with one stalled ribosome located at the initiation codon and then say: “An increase in EJC particle size well above 80S indicates that more than one ribosome remains stalled on the transcripts”. So how would you then interpret your data with harringtonine treated cells? You would expect only EJC-associated mRNPs that migrate into the gradient around the 80S peak but not further down.

As we now write in page 12: “ In the presence of harringtonine, elongating ribosomes are expected to terminate normally and drop off. Reinitiating ribosomes arrest as 80S particles. Therefore an EJC particle size remaining even increasing well above 80S, demonstrates that its ribosomes are not elongating translation, they are stalled. Both cycloheximide and harringtonine may favour reinitiation adding an extra ribosome to EJC particles ³¹. The increased in EJC particle size would be due to additional loading of ribosomes stalling at the initiation codon (harringtonine) or are frozen after initiation (cycloheximide).”

P. 13: Wouldn't a 4th and probably most obvious alternative explanation for the poor EJC enrichment on housekeeping genes be that they are undergoing very efficient translation, as soon as they enter the cytoplasm, which together with the notion that they tend to have a long half-life, would result in a low proportion of EJC-bound transcripts at any given time?

Indeed, there is a weak correlation between the amount of transcripts in inputs and bound to the EJC (Ejc0) (Figure 5d). The highest expressed transcripts are the weaker enriched to EJC (New Figure 5c). Highly abundant transcripts in inputs results in a high background (blk) immunoprecipitation hence low enrichment. As now shown in a new

panel (Figure 5b), enrichment in housekeeping gene transcripts remains significantly higher than enrichment of mitochondrial transcripts that do not undergo nuclear splicing hence EJC deposition.

In the last sentence of the Discussion, the authors finally speculate that the transcripts exhibiting persistent EJC enrichment might be those in which translation is delayed and dependent on a regulatory signal and/or their transport to the right locale in the cell. I find this an obvious hypothesis that could and should be tested by comparing the RIP-seq data with ribosome profiling data (see comment above).

Transcript abundance in “inputs” corresponds to their abundance in pooled 80S and polysome fractions. Thus it likely corresponds to a RNAs associated with ribosomes. This been said, ribosome footprinting should be and will be performed in future studies to investigate the positioning of stalled ribosomes on EJC-bound transcripts. However, ribosome profiling following immunoprecipitation of EJC-bound transcripts is experimentally challenging and it will most likely take time.

Minor points:

Legend to Fig. 2: Please improve clarity. Dotted and solid lines can hardly be distinguished (panels a, d, and e) and the sentence “RNA in 2A is detected by SYBR green fluorescence, bit in 2a, 2d and 2e ...” does not make sense. Please correct it.

OK

P. 6: A brief description of the assay would facilitate the reader to understand that the indicated times refer to the times during which the cells were incubated with the drugs before they were lysed and NanoBit activity was measured.

OK

Fig. 4c: Translate “ActD ou DRB” into English.

OK

RIP-seq: Please indicate in Fig. 5a which fractions from the sucrose gradient were pooled for the RIP-seq.

OK

P. 9: Please correct: “We could not find detect any straightforward ...”

OK

Figs. 6a and 6b should be swapped since panel b is referred to in the text earlier than panel a.

OK

P. 10: Please clarify the following sentence: “Overall, genes that are most affected by transcriptional arrest of 2 hours are the most affected by 1 hour.” Affected by 1 hour of what?

OK

P. 10: “neighboring genes genes immunoprecipitated ...”, delete one of the “genes”.

OK

P. 11, 2nd paragraph: “Therefore” instead of “Therefre”

OK

REVIEWER COMMENTS

Reviewer #1 (Remarks to the Author):

The authors performed a thorough revision and have addressed the minor concerns and suggestions that I raised during the first round of review. Thus, I remain very enthusiastic about this study. I think that is an important paper, not only for the NMD field but also for those interested in gene expression, in general. I do recommend publication in Nature commun.

Reviewer #2 (Remarks to the Author):

The authors have not responded in a meaningful way to my most important comment.

I suggested that they are underestimating the importance of fast EJC remodeling and requested some numerical modeling that was not performed. In their reply they continue to argue that after 1 hour of transcription arrest from steady state 50% of the EJCs remain, but this does not reflect what happens to the typical newly deposited EJC. For stable EJCs (with a half-life of 200 minutes) the steady state pool of EJCs reflect all of the EJCs deposited over multiple hours. For unstable EJCs (half-life of 12 minutes) only recently deposited EJCs are in the steady state pool. Therefore the steady state pool does not reflect the fate of a typical EJCs. Because of this I can not get the conclusions of the authors to work with any model.

If indeed, a newly deposited EJC had a 70% chance of being stable and a 30% chance of being unstable the steady state pool would consist of approximately 98% of stable EJCs and 2% of unstable ones. If one would measure decay from steady state, one would essentially detect only the stable pool, which is not what the authors report.

The ONLY way I can generate a model that produces graphs that resemble the author's is with half-lives of 12 and 200 minutes for the unstable and stable EJC, respectively, and relative deposition rates of 0.01692 and 0.00245 min⁻¹ for the unstable and stable EJC, respectively. Thus in the same time 1 stable EJC gets deposited, 7 unstable EJCs get deposited to get to the 70/30 steady state ratio. Thus, the vast majority of EJCs disappear rapidly after synthesis in a translation-dependent manner (with a half-life of 12 minutes), and only approximately 1 in 8 of deposited EJCs is stable (with a half-life of 200 minutes).

Not only does this make sense with the data in this manuscript, it also makes biological sense with what we know about EJs.

To explain it in the same terminology the authors use on p6:

At steady state assembly of equals disassembly rate:

For the stable EJs (1) $K_{assslow} = k_{offslow} * [EJC_{slow}]$

And for the unstable EJs (2) $K_{assfast} = k_{offfast} * [EJC_{fast}]$

Dividing equation 1 by 2 gives equation (3): $K_{assslow}/K_{assfast} = (k_{offslow} * [EJC_{slow}]) / (k_{offfast} * [EJC_{fast}])$

From the 200 and 12 minute half-lives we learn that the unstable EJs are disassembled 16.7 times faster (200/12) so (4) $k_{offfast} = 16.7 * k_{offslow}$

The authors show that at steady state (t=0) 70% of the EJs are of the stable kind, or (5) $[EJC_{slow}] = (70/30) * [EJC_{fast}] = 2.33 [EJC_{fast}]$

Substituting eq 4 and 5 into 3, we get (6) $K_{assslow}/K_{assfast} = (k_{offslow} * 2.33 [EJC_{fast}]) / (16.7 * k_{offslow} * [EJC_{fast}])$

Simplifying (6) becomes $K_{assslow}/K_{assfast} = 2.33/16.7 = 0.14$.

So for every 8 assembled EJs. 1 is slowly-disassembled and 7 are rapidly disassembled. Thus the author's data show that the vast majority of newly deposited EJs will be rapidly disassembled.

Minor point

In one place on p5 RNases should be RNases. This is spelled correctly elsewhere

Reviewer #3 (Remarks to the Author):

The authors have revised their manuscript according to the reviewers's suggestions and thereby addressed most of the raised points and solved previous misunderstandings or unclarities. However, the following points among my initial comments remain unclear and should be further clarified:

– I do not fully understand the authors' response to my question about why most of the NanoBiT signal still co-migrates with particles larger than 80S particles after Harringtonine treatment (Fig. 2e). The authors now changed the manuscript text to: "As both cycloheximide and harringtonine can favour reinitiation adding an extra ribosome to EJC particles 32, the observed increased EJC particle size might be attributed to additional loading of a ribosome". I find this statement difficult to understand. Where on the RNA would this postulated additional ribosome be located and how would it get there in Harringtonine-treated cells? How does this reported reinitiation (ref 32) work? The authors should explain better how they interpret the observed NanoBiT signal distribution in the sucrose gradient fractions.

– The first paragraph added to the section "Increasing ionic strength relieves constraints in EJC-ribosome particles" attempts to explain how the increase of ionic strength and the RNase treatment is expected to affect mRNPs. That is laudable. However, the explanation is a bit convoluted, and the text doesn't flow well. The authors should improve the clarity of this paragraph by rephrasing it.

– Reviewer #2 apparently had similar concerns as I (reviewer #3) had regarding the analysis and interpretation of the data obtained from the complex time course experiments shown in Fig. 4, which led to the conclusion that only 30% of the EJCs are disassembled from RNA rapidly and in a translation-dependent manner. I still find it difficult to reconcile the data of Fig. 4 a-d with the data of Fig. 4e. I encourage the authors to repeat their calculations to estimate the percentage of translation-independent EJC disassembly under the assumption that the data in Fig. 4e would be accurate and the data in Fig. 4a-d would be less trustworthy. This exercise will show how much the calculated percentage of translation-dependent EJC disassembly changes based on which experimental data is considered and give a feeling about the confidence we can have in this number. In my opinion, the unexpectedly low percentage of translation-dependent EJC disassembly is a key conclusion of the paper, presumably the main conclusion that will be associated with this work, and it should therefore be as reliable and robust as possible.

Trivia:

– I think there is a mistake in the sentence added to figure legend 1: Data from biological triplicates is shown in d-g, not b-e.

– Correct the typo on page 5: alòòthough

Once again we thank the reviewers for their useful comments, in particular their comments on the kinetic analysis of EJC dissociation. To take into account the comments, the abstract, paragraphs in the result and discussion sections, have been rephrased. Furthermore, schemes have been added to Figures 2 and 4.

Reviewer #1 (Remarks to the Author):

The authors performed a thorough revision and have addressed the minor concerns and suggestions that I raised during the first round of review. Thus, I remain very enthusiastic about this study. I think that is an important paper, not only for the NMD field but also for those interested in gene expression, in general. I do recommend publication in Nature commun.

Reviewer #2 (Remarks to the Author):

The authors have not responded in a meaningful way to my most important comment.

I suggested that they are underestimating the importance of fast EJC remodeling and requested some numerical modeling that was not performed. In their reply they continue to argue that after 1 hour of transcription arrest from steady state 50% of the EJCs remain, but this does not reflect what happens to the typical newly deposited EJC. For stable EJCs (with a half-life of 200 minutes) the steady state pool of EJCs reflect all of the EJCs deposited over multiple hours. For unstable EJCs (half-life of 12 minutes) only recently deposited EJCs are in the steady state pool. Therefore the steady state pool does not reflect the fate of a typical EJCs. Because of this I can not get the conclusions of the authors to work with any model.

If indeed, a newly deposited EJC had a 70% chance of being stable and a 30% chance of being unstable the steady state pool would consist of approximately 98% of stable EJCs and 2% of unstable ones. If one would measure decay from steady state, one would essentially detect only the stable pool, which is not what the authors report.

The ONLY way I can generate a model that produces graphs that resemble the author's is with half-lives of 12 and 200 minutes for the unstable and stable EJC, respectively, and relative deposition rates of 0.01692 and 0.00245 min⁻¹ for the unstable and stable EJC, respectively. Thus in the same time 1 stable EJC gets deposited, 7 unstable EJCs get deposited to get to the 70/30 steady state ratio. Thus, the vast majority of EJCs disappear rapidly after synthesis in a translation-dependent manner (with a half-life of 12 minutes), and only approximately 1 in 8 of deposited EJCs is stable (with a half-life of 200 minutes). Not only does this make sense with the data in this manuscript, it also makes biological sense with what we know about EJCs.

To explain it in the same terminology the authors use on p6:
At steady state assembly of equals disassembly rate:
For the stable EJCs (1) $K_{assslow} = k_{offslow} * [EJC_{0slow}]$
And fro the unstable EJCs (2) $K_{assfast} = k_{offfast} * [EJC_{0fast}]$

Dividing equation 1 by 2 gives equation (3): $K_{assslow}/K_{assfast} = (k_{offslow}*[EJC_{0slow}]) / (k_{offfast}*[EJC_{0fast}])$

From the 200 and 12 minute half-lives we learn that the unstable EJCs are disassembled 16.7 times faster (200/12) so (4) $K_{offfast} = 16.7*K_{offslow}$

The authors show that at steady state (t=0) 70% of the EJCs are of the stable kind, or (5) $[EJC_{0slow}] = (70/30)* [EJC_{0fast}] = 2.33 [EJC_{0fast}]$

Substituting eq 4 and 5 into 3, we get (6) $K_{assslow}/K_{assfast} = (k_{offslow}*2.33 [EJC_{0fast}]) / (16.7*K_{offslow} *[EJC_{0fast}])$

Simplifying (6) becomes $K_{assslow}/K_{assfast}=2.33/16.7= 0.14$.

So for every 8 assembled EJCs. 1 is slowly-disassembled and 7 are rapidly disassembled. Thus the author's data show that the vast majority of newly deposited EJCs will be rapidly disassembled.

We are indebted to reviewer 2 for insisting in raising this issue. We did not imagine it was possible to discuss the assembly process. Indeed we agree that although only 30% of the EJCs are disassembled slowly independently of translation after an arrest in transcription, paradoxically at steady-state 7 out of 8 newly assembled EJCs are disassembled by the fast mechanism. We have added a new paragraph in the result section (**Translation-dependent EJC assembly and disassembly are fast processes**), added a scheme as panel (e) in Figure 4 and modified the abstract accordingly. Furthermore, we revised our formalism distinguishing rates (R) and rate constants (k).

“Now, one might want to question the probability of a newly deposited EJC to be disassembled by a translation-dependent (TD) or translation-independent (TinD) mechanism. The respective contributions of these pathways may be estimated comparing steady state levels and assuming that disassembly rates are proportional to EJC concentrations (EJC). At steady state, the EJC concentration (EJC^0) is constant over time, the net reaction rate is zero, assembly rates (R_{ass}) are equal to disassembly rates (R_{diss}),.

At steady state,

$$R_{ass}^{TinD} = R_{diss}^{TinD} = k_{diss}^{TinD} (EJC^{0-TinD}) \text{ and } R_{ass}^{TD} = R_{diss}^{TD} = k_{diss}^{TD} (EJC^{0-TD})$$

Assuming that one distinguishes two EJC populations, those that are disassembled slowly independently of translation (EJC^{TinD}) from those that are disassembled rapidly and translation dependent (EJC^{TD}).

$$R_{ass}^{TD} / R_{ass}^{TinD} = R_{diss}^{TD} / R_{diss}^{TinD} = (k_{diss}^{TD} / k_{diss}^{TinD})((EJC^{0-TD})/(EJC^{0-TinD}))$$

According to the data provided in Figure 4d $(EJC^{0-TD})/(EJC^{0-TinD}) = 30/70$ and

$$(k_{diss}^{TD} / k_{diss}^{TinD}) = \tau_1/\tau_2 = 17$$

Therefore:

$$R_{ass}^{TD} / R_{ass}^{TinD} \approx 17 \times (30/70) = 7.$$

The assembly rate for EJCs that undergo fast translation-independent (TD) disassembly is 7-fold faster than that of those that undergo fast translation-dependent (TD) disassembly. Steady state concentrations are the result of opposed assembly and disassembly processes. Although their assembly is 7-fold faster, “unstable” EJCs

accumulate less than “stable” ones because their dissociation is much faster (17-fold) than that of “unstable” ones (Figure 4e). Paradoxically at steady state most, 7 out of 8 newly deposited EJCs are disassembled by a fast translation-dependent process. EJC removal from mRNA had already been proposed to follow both translation-dependent and translation-independent mechanisms¹⁹. We now provide an estimate of their respective contributions. “

In the discussion section we write: “Translating ribosomes account for a minor proportion of EJC disassembly after treatment with inhibitors. However paradoxically, at steady state most, 85% of newly assembled EJCs are disassembled by a fast, translation-dependent process in agreement with the common thought that EJCs are removed by the pioneer round of translation.”

While re-examining the kinetic data, we realized that increases in EJC concentrations upon addition of translation inhibition were not consistent with the kinetic data. They were too small. As one cannot but speculate for the reasons, this aspect of the analysis was deleted. Noteworthy, there is a good parallel between kinetics following addition of cycloheximide to untreated (now Figure 4c) or DRB-treated (now Figure 4f) cells.

Minor point

In one place on p5 RNases should be RNases. This is spelled correctly elsewhere
OK

Reviewer #3 (Remarks to the Author):

The authors have revised their manuscript according to the reviewers’s suggestions and thereby addressed most of the raised points and solved previous misunderstandings or unclarities. However, the following points among my initial comments remain unclear and should be further clarified:

1) – I do not fully understand the authors’ response to my question about why most of the NanoBiT signal still co-migrates with particles larger than 80S particles after Harringtonine treatment (Fig. 2e). The authors now changed the manuscript text to: “As both cycloheximide and harringtonine can favour reinitiation adding an extra ribosome to EJC particles³², the observed increased EJC particle size might be attributed to additional loading of a ribosome”. I find this statement difficult to understand. Where on the RNA would this postulated additional ribosome be located and how would it get there in Harringtonine-treated cells? How does this reported reinitiation (ref 32) work? The authors should explain better how they interpret the observed NanoBiT signal distribution in the sucrose gradient fractions.

We removed this discussion from the result section, added an explicative scheme (Figure 2 – panel f) and rephrased the following paragraph in the discussion section :

“In the presence of harringtonine, elongating ribosomes are expected to terminate normally and drop off. Reinitiating ribosomes arrest at the start codon as 80S particles. Therefore an EJC particle size remaining well above 80S, demonstrates that its ribosomes do not elongate translation and drop off. They are stalled. On the contrary, an

increased EJC particle size is observed upon inhibition of translation. Both cycloheximide and harringtonine treatments can lead to queuing of preinitiation complexes and enhance recognition of weak non-AUG start codons³². Hence the increased EJC particle size might be attributed to the additional loading of ribosomes arrested at weak non-canonical initiation codon (harringtonine) or frozen just after initiation (cycloheximide) and preribosomes queuing upstream (Figure 2f). “

2) – The first paragraph added to the section “Increasing ionic strength relieves constraints in EJC-ribosome particles” attempts to explain how the increase of ionic strength and the RNase treatment is expected to affect mRNPs. That is laudable. However, the explanation is a bit convoluted, and the text doesn’t flow well. The authors should improve the clarity of this paragraph by rephrasing it.

We have rephrased the paragraph “Increasing ionic strength relieves constraints in EJC-ribosome particles” in the result section to keep it descriptive.

In the discussion section, we now write :

“Compaction of the EJC-ribosome particles might limit access to the nucleases between ribosomes and EJC and also restrict the EJC-NanoBiT activity. Increasing ionic strength loosens protein/protein and protein/RNA interactions leading to a partial release of the constraints likely responsible for the increased accessibility to RNase accessibility to RNase. Thus, high ionic strength is required for an RNase treatment to completely detach the EJC from ribosomes. In addition, luciferase activity increases markedly upon RNase digestion at high ionic strength. The split luciferase fragments should not be prevented to interact by neighboring macromolecules. The EJC-luciferase gives a global view, some EJC split fragments under constraint might not be able to interact at all whereas others may. As ionic strength and RNase digestion weaken protein interactions, release the ribosomes and therefore removes constraints preventing efficient split-luciferase matching.”

3) – Reviewer #2 apparently had similar concerns as I (reviewer #3) had regarding the analysis and interpretation of the data obtained from the complex time course experiments shown in Fig. 4, which led to the conclusion that only 30% of the EJCs are disassembled from RNA rapidly and in a translation-dependent manner.

We have added a full paragraph in the result section, developing the arguments put forward by reviewer 2 : Although only 30% of EJCs present at steady state undergo translation-dependent EJC disassembly, paradoxically, 7 out of 8 EJCs deposited at steady-state undergo translation-dependent EJC disassembly. A scheme (new Figure 4e) illustrates this idea.

I still find it difficult to reconcile the data of Fig. 4 a-d with the data of Fig. 4e. I encourage the authors to repeat their calculations to estimate the percentage of translation-independent EJC disassembly under the assumption that the data in Fig. 4e would be accurate and the data in Fig. 4a-d would be less trustworthy. This exercise will show how much the calculated percentage of translation-dependent EJC disassembly changes based on which experimental data is considered and give a feeling about the

confidence we can have in this number. In my opinion, the unexpectedly low percentage of translation-dependent EJC disassembly is a key conclusion of the paper, presumably the main conclusion that will be associated with this work, and it should therefore be as reliable and robust as possible.

We have fused in the same paragraph **“Observing assembly of the EJC following resumption of transcription”** the analysis of translation inhibition on untreated and DRB treated cells.

We have made more quantitative estimate of recovery rate constants now given in Figure 4f (former Figure 4e). We performed, a simple numerical analysis to estimate the rates when the recovery has generated 0.75 (EJC⁰), starting from 0.5 (EJC⁰). In these conditions, we obtain a recovery rate in the presence of cycloheximide 3.4 faster than without which compares well with the experimental data (Calculations at the end of this document). A rigorous mathematical treatment is quite tricky and not justified given experimental uncertainties and poor knowledge in transcription/splicing recovery delays following inhibition of transcription. Nevertheless, the point we want to make is that new EJCs are assembled when transcription resumes and this process does not require protein neo-synthesis. It illustrates the common view that transcription is a requirement for EJC assembly.

We now write as a description of Figure 4f (recovery) :

“Attempting to investigate EJC assembly directly, we followed its formation after removal of transcription inhibition. DRB is reversible and well adapted to transcription recovery studies and has successfully been used to evaluate rates of transcription and splicing in live human cells³³. Thus, after 2h of treatment the DRB is washed away and replaced with fresh medium. The luciferase activity then increases back rapidly (Figure 4f). If DRB-containing medium is replaced by cycloheximide-containing medium, the recovery is 3.5-fold faster globally during the first hour and plateaus. A rigorous mathematical treatment is quite tricky and not justified given experimental uncertainties and poor knowledge in transcription/splicing recovery delays following inhibition of transcription. Furthermore, transcriptional stress is released, specific transcripts might be transcribed and these transcripts might be less susceptible than steady state ones to fast translation-dependent EJC disassembly. We performed a simple numerical analysis (not shown) using the numbers from Figure 4c to estimate the rates when the recovery has generated 0.75 (EJC⁰), starting from 0.5 (EJC⁰). In these conditions, we obtain a recovery rate in the presence of cycloheximide 3.4 faster than without which compares well with the experimental data. Noteworthy, increases in EJC levels in the presence of cycloheximide occur at approximately the same rates when cycloheximide is added to untreated (Figure 4e) or DRB treated cells (Figure 4f). The recovery is faster in the presence of cycloheximide in consistency with significant contribution of a translation-dependent disassembly. This experiment shows that new EJCs are assembled as soon as transcription resumes. EJC assembly involves transcription but this process does not require protein neo-synthesis. It illustrates the common view that transcription is a requirement for EJC assembly.

”

Trivia:

- I think there is a mistake in the sentence added to figure legend 1: Data from biological triplicates is shown in d-g, not b-e.

OK

- Correct the typo on page 5:

alòòthough

OK

KINETIC ANALYSIS of RECOVERY AFTER DRB

At steady state

$$d(\text{EJC})/dt = R_{\text{ass}}^{\text{TD}} + R_{\text{ass}}^{\text{TinD}} - R_{\text{diss}}^{0\text{-TD}} - R_{\text{diss}}^{0\text{-TinD}} = 0$$

$$R_{\text{ass}}^{\text{TD}} + R_{\text{ass}}^{\text{TinD}} = R_{\text{diss}}^{0\text{-TD}} + R_{\text{diss}}^{0\text{-TinD}}$$

$$R_{\text{diss}}^{0\text{-TinD}} = k_{\text{diss}}^{\text{TinD}} (\text{EJC}^{0\text{-TinD}}) \text{ and } R_{\text{diss}}^{0\text{-TD}} = k_{\text{diss}}^{\text{TD}} (\text{EJC}^{0\text{TD}})$$

$$\text{As } (\text{EJC}^{\text{TD}}) = 0.3 (\text{EJC}^0); (\text{EJC}^{\text{TinD}}) = 0.7 (\text{EJC}^0); k_{\text{diss}}^{\text{TD}} = 17 k_{\text{diss}}^{\text{TinD}}$$

$$R_{\text{diss}}^{0\text{-TD}} = 0.3 k_{\text{diss}}^{\text{TD}} (\text{EJC}^0)$$

$$R_{\text{diss}}^{0\text{-TD}} + R_{\text{diss}}^{0\text{-TinD}} = 17 \times 0.3 k_{\text{diss}}^{\text{TinD}} (\text{EJC}^0) + 0.7 k_{\text{diss}}^{\text{TinD}} (\text{EJC}^0)$$

$$R_{\text{ass}}^{\text{TD}} + R_{\text{ass}}^{\text{TinD}} = 5.4 k_{\text{diss}}^{\text{TinD}} (\text{EJC}^0)$$

In the presence of DRB

$$R_{\text{ass}}^{\text{TD}} = R_{\text{ass}}^{\text{TinD}} = 0$$

$$d(\text{EJC})/dt = - R_{\text{diss}}^{0\text{-TD}} - R_{\text{diss}}^{0\text{-TinD}}$$

$$d(\text{EJC})/dt = - k_{\text{diss}}^{\text{TD}} (\text{EJC}^{\text{TD}}) - k_{\text{diss}}^{\text{TinD}} (\text{EJC}^{\text{TinD}}) = - 17 k_{\text{diss}}^{\text{TD}} (\text{EJC}^{\text{TD}}) - k_{\text{diss}}^{\text{TinD}} (\text{EJC}^{\text{TinD}})$$

Rates upon removal of DRB

$$d(\text{EJC})/dt = R_{\text{ass}}^{\text{TD}} + R_{\text{ass}}^{\text{TinD}} - R_{\text{diss}}^{\text{TD}} - R_{\text{diss}}^{\text{TinD}}$$

We assume that $R_{\text{ass}}^{\text{TD}}$ and $R_{\text{ass}}^{\text{TinD}}$ remain the same that at steady state

For dissociation rates, we make calculations for $(\text{EJC}) = 0.5 (\text{EJC}^0)$ after 2H00 in DRB and $(\text{EJC}) = 0.75 (\text{EJC}^0)$ after 30' recovery (Figure 4f)

Since assembly is 7 fold higher for translation-dependent than independent EJCs and $(\text{EJC}^{\text{TD}}) = 0$ when DRB is removed.

$$(\text{EJC}^{\text{TD}}) = 0.21 (\text{EJC}^0) \text{ and } (\text{EJC}^{\text{TinD}}) = 0.54 (\text{EJC}^0)$$

$$R_{\text{diss}}^{\text{TD}} + R_{\text{diss}}^{\text{TinD}} = 0.21 \times 17 k_{\text{diss}}^{\text{TinD}} (\text{EJC}^0) + 0.54 k_{\text{diss}}^{\text{TinD}} (\text{EJC}^0) = 4.1 k_{\text{diss}}^{\text{TinD}} (\text{EJC}^0)$$

$$d(\text{EJC})/dt = 5.4 k_{\text{diss}}^{\text{TinD}} (\text{EJC}^0) - 4.1 k_{\text{diss}}^{\text{TinD}} (\text{EJC}^0) = 1.3 k_{\text{diss}}^{\text{TinD}} (\text{EJC}^0)$$

Rates upon removal of DRB in the presence of cycloheximide

$$d(\text{EJC})/dt = R_{\text{ass}}^{\text{TD}} + R_{\text{ass}}^{\text{TinD}} - R_{\text{diss}}^{\text{TinD}} \quad \text{because} \quad R_{\text{diss}}^{\text{TD}} = 0$$

For dissociation rates, we make calculations for (EJC) = 0.5 (EJC⁰) after 2H00 in DRB and (EJC) = 0.75 (EJC⁰) after 10' recovery (Figure 4f)

$$d(\text{EJC})/dt = 5.4 k_{\text{diss}}^{\text{TinD}} (\text{EJC}^0) - 0.75 k_{\text{diss}}^{\text{TinD}} (\text{EJC}^0) = 4.45 k_{\text{diss}}^{\text{TinD}} (\text{EJC}^0)$$

$$\text{Ratio} - (\text{Rate} + \text{CHX}) / (\text{Rate without}) = 4.45 / 1.3 = 3.4$$

REVIEWERS' COMMENTS

Reviewer #2 (Remarks to the Author):

The authors have fully addressed my previous concerns and the conclusions are now well supported by the data. The manuscript is suitable for publication in Nature Communications.

Reviewer #3 (Remarks to the Author):

Thanks to reviewer 2's precise advice, the initial misinterpretation of the kinetic data has been partly solved. The authors now arrive at the conclusion that 85% of the assembled EJCs are short-lived and undergo translation-dependent disassembly, in agreement with previous reports.

However, I don't understand why the authors call this finding "paradoxical" and why they still state in the abstract that 70% of the EJCs present at a given time at steady state persist several hours and undergo translation-independent disassembly. This is incorrect and misleading, because their measurements in the transcription-inhibited situation do not represent the situation in unperturbed cells as has been well explained by reviewer 2: In their experiment, all stable EJCs deposited over multiple hours are present, whereas only recently deposited unstable EJCs are present, leading to a strong overrepresentation of the stable EJC fraction.

In their revised version of the manuscript, the authors have now added the correct kinetic analysis and interpretation, but have not yet removed the wrong one, leading them to talk about paradoxical findings. Instead, they should just remove the incorrect interpretation (that 70% of EJC were stable and disassembled independent of translation) and the paradox would disappear.

The other two points have been clarified

POINT BY POINT RESPONSE TO REVIEWER 3's COMMENTS

Reviewer #3 (Remarks to the Author):

Thanks to reviewer 2's precise advice, the initial misinterpretation of the kinetic data has been partly solved. The authors now arrive at the conclusion that 85% of the assembled EJCs are short-lived and undergo translation-dependent disassembly, in agreement with previous reports.

However, I don't understand why the authors call this finding "paradoxical" and why they still state in the abstract that 70% of the EJCs present at a given time at steady state persist several hours and undergo translation-independent disassembly. This is incorrect and misleading, because their measurements in the transcription-inhibited situation do not represent the situation in unperturbed cells as has been well explained by reviewer 2: In their experiment, all stable EJCs deposited over multiple hours are present, whereas only recently deposited unstable EJCs are present, leading to a strong overrepresentation of the stable EJC fraction. In their revised version of the manuscript, the authors have now added the correct kinetic analysis and interpretation, but have not yet removed the wrong one, leading them to talk about paradoxical findings. Instead, they should just remove the incorrect interpretation (that 70% of EJC were stable and disassembled independent of translation) and the paradox would disappear.

The other two points have been clarified

Indeed, 70% of the EJCs present at a given time at steady state persist several hours and undergo translation-independent disassembly. This is not an artifact due to the presence of an inhibitor because it is an extrapolation of curves at time 0 when there is no inhibitor and therefore is untreated. Yet most (85%) of the newly deposited EJCs undergo translation-dependent disassembly. Following reviewer 2's suggestions we show that this apparent discrepancy or paradox is accounted for by a model in which EJCs prone for translation-dependent disassembly assemble faster (7-fold) than those EJCs prone for translation-dependent disassembly but the disassembly of the translation-dependent is much faster (17-fold) (Scheme Figure 4d). We have added the numerical analysis of EJC recovery following DRB removal (to model data in figure 4f) as Supplementary method 1, we made a few changes in the abstract and the discussion in an attempt to make this point more clear.